**Impact of diurnal temperature fluctuations on larval settlement and growth of the reef coral *Pocillopora damicornis***

Lei Jiang[1, 2, 3], You-Fang Sun[1, 2, 3], Yu-Yang Zhang[1], Guo-Wei Zhou[1, 2], Xiu-Bao Li[1],

Laurence J. McCook[1, 4], Jian-Sheng Lian[1], Xin-Ming Lei[1], Sheng Liu[1], Lin Cai[5],

Pei-Yuan Qian[5, *], Hui Huang[1, 2, *]

Key Laboratory of Tropical Marine Bio-resources and Ecology, Guangdong

Provincial Key Laboratory of Applied Marine Biology, South China Sea Institute of

Oceanology, Chinese Academy of Sciences, Guangzhou 510301, China

Tropical Marine Biological Research Station in Hainan, Chinese Academy of

Sciences, Sanya 572000, China

University of Chinese Academy of Sciences, Beijing 100049, China
ARC Centre of Excellence for Coral Reef Studies, James Cook University,

Townsville, Australia

Shenzhen Research Institute and Division of Life Science, Hong Kong University of

Science and Technology, Hong Kong SAR, China

**\* Corresponding authors**:

Pei-Yuan Qian, tel. +85 223 587 331, fax +85 223 581 559, e-mail: boqianpy@ust.hk

Hui Huang, tel & fax +86 284 460 294, e-mail: huanghui@scsio.ac.cn

**Abstract**

Diurnal fluctuations in seawater temperature are ubiquitous on tropical reef flats. However, the effects of such dynamic temperature variations on the early stages of corals are poorly understood. In this study, we investigated the responses of larvae and new recruits of *Pocillopora damicornis* to two constant temperature treatments (29 and 31 ℃), and two diurnally fluctuating treatments (28–31 and 30–33 ℃ with daily means of 29 and 31 ℃, respectively) simulating the 3 ℃ diel oscillations at 3 m depth on Luhuitou fringing reef (Sanya, China). Results showed that the thermal stress on settlement at 31 ℃ was almost negated by the fluctuating treatment. Further, neither elevated temperature nor temperature fluctuations caused bleaching responses in recruits, while the maximum excitation pressure over photosystem II (PSII) was reduced under fluctuating temperatures. Although early growth and development were highly stimulated at 31 ℃, oscillations of 3 ℃ had little effects on budding and lateral growth at either mean temperature. Nevertheless, daytime encounters with the maximum temperature of 33 ℃ in fluctuating 31 ℃ elicited a notable reduction in calcification compared to constant 31 ℃. These results underscore the complexity of the effects caused by diel temperature fluctuations on early stages of corals, and suggest that ecologically relevant temperature variability could buffer warming stress on larval settlement and dampen the positive effects of increased temperatures on coral growth.

**Keywords**: temperature, diurnal fluctuation, *Pocillopora damicornis*, settlement, bleaching, calcification, budding

**1 Introduction**

Scleractinian corals and the reef ecosystems they construct are currently facing environmental changes at unprecedented rates of changes. Of these changes, rising seawater temperature is generally recognized as one of the most immediate and widespread threats (Hoegh-Guldberg, 1999; Hughes et al., 2003). The most conspicuous response of corals to elevated temperatures is to expel their endosymbiotic dinoflagellates and/or photosynthetic pigments, giving the affected colonies a pale appearance, a process known as coral bleaching (Hoegh-Guldberg, 1999). Due to the loss of zooxanthellae, bleached corals usually fail to obtain their key metabolic requirements from photosynthetically fixed carbon (Grottoli et al., 2006). As a result, massive mortality of corals has been frequently observed following bleaching, leading to serious decline and impaired ecosystem functionality (Hoegh-Guldberg, 2011; Graham et al., 2006).

On average, sea surface temperatures have increased by approximately 0.7 ℃ since preindustrial times (Feely et al., 2013) and a further increase of 2–3 ℃ is expected by the end of this century (Bopp et al., 2013), giving rise to increased concerns about effects on corals. The bulk of scientific work addressing the impact of ocean warming on corals has focused on their tolerance and physiological responses to the predicted increases in mean temperature (Stambler, 2010). However, seawater temperatures ~~on coral reefs~~ are characterized by ~~striking~~ fluctuations over timescales ranging from minutes to hours to months. Notably, temperature profiles from reef environments typically show diel oscillations of 4–10 ℃ (Coles, 1997; Dandan et al., 2015;

Guadayol et al., 2014; Oliver and Palumbi, 2011; Rivest and Gouhier, 2015). A
consistent daily cycle is commonly present, with temperature increasing after sunrise,
peaking after noon and then gradually decreasing to the minimum (e.g., Zhang et al.,
2013; Putnam and Edmunds, 2011).
It has been long established that the performance of organisms, including a diverse
range of marine invertebrates, differs between steady and variable thermal conditions
~~of~~ at equivalent mean temperature (Bryars and Havenhand, 2006; Lucas and Costlow,
1979; Marshall and McQuaid, 2010; Orcutt and Porter, 1983; Pilditch and Grant, 1999;
Sastry, 1979). These studies have demonstrated that temperature fluctuations can
either speed up or retard early development and growth, depending upon the mean
temperatures and amplitude of the fluctuations. However, few studies have explored
this thermodynamic effect on corals which routinely experience temperature
oscillations in nature (e.g., Mayfield et al., 2012; Putnam et al., 2010).
Recently, our understanding of the physiological responses of corals to diurnally
fluctuating temperature has advanced, but results have been variable and even
conflicting. For instance, the photo-physiology in larval and adult pocilloporid corals
is more ~~suited to the fluctuating than to the constant~~ adapted to fluctuating
temperatures (Mayfield et al., 2012; Putnam et al., 2010). Conversely, ~~evidence for~~
~~the deleterious effects of diel temperature fluctuations includes the~~ significant
reductions in photochemical efficiency, symbiont density and aerobic respiration were
found in corals exposed to fluctuating temperatures compared to those in constant
temperatures (Putnam and Edmunds, 2011; Putnam and Edmunds, 2008). These
contrasting results emphasize a clear need to further explore the impact of diurnally
fluctuating temperatures, together with the projected increase in temperature on reef
corals.

In the context of global deterioration of coral reefs and climate change, the early

life history stages of corals have drawn increasing attention in recent decades, as they
are more vulnerable to environmental changes than their adult counterparts, and more
importantly, represent a bottleneck for the maintenance of populations (Byrne, 2012;
Keshavmurthy et al., 2014). Successful larval settlement, post-settlement survival and
growth are of paramount importance to population persistence, as well as the recovery
of degraded reefs (Ritson-Williams et al., 2009; Penin and Adjeroud, 2013). Mounting
evidence suggests that ocean warming poses a serious threat to these early processes
(reviewed in Keshavmurthy et al., 2014), but most previous experiments utilized
steady temperature treatments, neglecting the temporal variations of *in situ*
temperature (but see Putnam et al., 2010). To date, there is a paucity of knowledge
regarding the influence of dynamic temperatures on these crucial early stages of reef
corals. The risk imposed by ocean warming on fitness and development of corals can
be best understood by integrating both diel thermocycles and changes in mean
temperature (Boyd et al., 2016).

The present study aimed to investigate how the early stages of the reef coral

*Pocillopora damicornis* will be affected by the diurnally oscillatory temperatures,
together with ocean warming. *P. damicornis* is a widely distributed and major
reef-building coral on reef flats in the Indo-Pacific region (Veron, 1993). This species

planulates almost every month and the release of free-swimming ~~and~~ zooxanthellate planula larvae follows a lunar cycle (Fan et al., 2002). Brooded larvae and new recruits were exposed to two temperature levels (29 and 31 ℃) crossed with two temperatures regimes (constant and 3 ℃ diel fluctuations). Diurnal patterns of temperature fluctuations were based on temperature records from our study site, Luhuitou fringing reef in Sanya, China. Larval condition and juvenile growth after incubation were assessed to compare their responses to constant and oscillatory temperatures.

## 2 Materials and methods

### 2.1 Field seawater temperature monitoring

Seawater temperatures at 3 m depth on Luhuitou fringing reef (18°12′N, 109°28′E) were recorded at 30 min intervals from 2012 to 2016, using Hobo Pendant data loggers (Onset, USA). The temperature profiles showed large seasonal and diurnal fluctuations, with a maximum of 33.1 ℃ and a minimum of 20.3 ℃ (Fig. S1a). The mean annual temperature was 27 ℃ and the mean monthly temperature ranged from 22 to 30.2 ℃ (Fig. S1b). The diurnal range in temperature variation during summer (June–September) was between 0.6 and 5.4 ℃, with a mean value of 1.76 ℃ (Fig. S1c). Each day, seawater temperature began to rise at ~~increase~~ around 08:00, reached the maximum at 13:00, often remained constant for about two hours, and then gradually decreased (Fig. S1d).

**2.2 Sampling of corals and ~~C~~larval ~~c~~ollection ~~and allocation of coral larvae~~**

Eight *P. damicornis* colonies were collected at a depth of 3 m on 20 August 2015. Colonies were transported to Tropical Marine Biological Research Station, and placed individually into 20 L flow-through tanks at ambient temperature (28.7 ± 0.5 ℃) under partially shaded light conditions (noon irradiance, ~300 μmol photons m$^{-2}$ s$^{-1}$). The outflow of each tank was passed through a cup fitted with 180 μm ~~mesh~~ net on the bottom to trap larvae. Larvae released from these colonies were collected at 07:00 on 22 August 2015~~,~~ and then pooled. ~~and~~ Two groups of larvae were haphazardly selected for~~assigned to~~ the following ~~two~~ settlement and recruit experiments, respectively. ~~For the settlement assays, larvae were introduced to 5.5 cm diameter plastic petri dishes as described below (see Section 2.4). To test the effects of temperature treatments on the photo-physiology and growth of recruits, another batch of larvae were transferred to 10-cm-diameter petri dishes which were left floating in a flow-through tank. Twenty hours later, 4 dishes with a total of 35–40 newly settled recruits were assigned to each treatment tank. Only recruits that settled individually and at least 1 cm apart from others were selected for the experiment in order to avoid possible contact between recruits during growth. Dishes were rotated daily to avoid the potential positional effects within each tank.~~

## 2.3 Experimental setup

The two temperature regimes, constant and fluctuating, were set for the target temperature levels of 29 ℃ and 31 ℃ each. The later temperature value ~~29 ℃ treatment, corresponding to the ambient temperature at the collection site of adult *P. damicornis*, was used as the control treatment. The experimental temperature~~ was 2 ℃ above the ambient and 1 ℃ above the bleaching threshold for coral communities on Luhuitou reef (30 ℃, Li et al., 2012), and within the range of projected increases (Bopp et al., 2013). ~~Two temperature regimes, i.e., constant and fluctuating were set for each temperature level.~~ The pattern and range of temperatures in the two fluctuating treatments were based on *in situ* records obtained during larval release of *P. damicornis* (Fig. S1d), and the assumption that the predicted 2 ℃ increase in mean temperature would entail a 2 ℃ shift in the overall temperature time-course (Burroughs, 2007). The 29 ℃ treatment, corresponding to the ambient temperature at the collection site of adult *P. damicornis*, was used as the control treatment.

All incubations were carried out in ~~Four~~ 40 L tanks which were filled with sand-filtered seawater~~, which~~. Seawater in each tank was partially changed (30%) with temperature-equilibrated seawater at 22:00 every day. Temperature regimes~~Treatments~~ were set using digital temperature regulators (Sieval, TC-05B, China) and 50 W heaters. The seawater was gently aerated and well mixed using submerged pumps (350 L h$^{-1}$). The water temperature in each tank was recorded with a Hobo Pendant logger at 15 min intervals throughout the experiment. In the two fluctuating treatments (Fig. 1), temperatures were programmed to increase from

28/30 ℃ at 08:00, reach the plateau of 31/33 ℃ around 13:00 and stabilize for 2
hours. At 15:00, temperatures were allowed to decrease gradually to 28/30 ℃ around
22:00 and remained stable until 09:00 the next morning. Mean (± SD) daily
temperature of the two stable treatments were 29 ± 0.2 and 30.8 ± 0.2 ℃, and the
mean temperatures of the two fluctuating treatments were 28.9 ± 1.3 and 30.7 ±
1.3 ℃ respectively. Salinity in each tank was checked using an Orion 013010MD
conductivity probe twice a day and remained stable at 33 psu during the experiment.
Each tank was illuminated by a LED lamp (Maxspect, 10,000K, China) on a 12:12
h light-dark cycle. Light was measured with a Li-Cor 4-$\pi$ quantum sensor below the
water surface. Light intensity was similar in all tanks ($F_{3, 96}$ = 0.32, $P$ = 0.81),
averaging at 183 ± 3 μmol photons m$^{-2}$ s$^{-1}$ (mean ± SE, n = 100), which was close to
the irradiance in crevices where coral recruits were found at 3−4 m depths at our study
site (Lei Jiang, unpublished data). Facility and logistical constraints precluded the
replication of treatments, but salinity and light were carefully controlled to eliminate
any possible artefact (Underwood, 1997).

**2.4 Larval sSettlement assay**
To explore the impact of temperature treatments on larval settlement, 240 larvae were
randomly selected for Tthe settlement experiments were . Settlement assays were
conducted in 5.5-cm-diameter petri dishes on 22 August 2015 and began starting at
around 09:00. The crustose coralline algae (CCA), *Hydrolithon reinboldii*, one of the

most abundant CCA species and an effective settlement cue for larval settlement of *P. damicornis* at our study site, was collected at 2–3 m depths and cut into uniformly sized (5 × 5 × 3 mm) chips 4 days before the settlement experiment. Each dish contained 15 ml seawater and a CCA chip. Fifteen actively swimming larvae were introduced into each dish, which was then floated and partially (80%) submerged in seawater ~~in the treatment tanks~~ to ensure temperature control. Preliminary measurements showed that the difference in seawater temperature between dishes and tanks was less than 0.4 ℃. Four replicate dishes were used for each treatment. Larvae were allowed to settle for 24 hours, after which settlement success was assessed under a dissecting microscope following the criteria of Heyward and Negri (1999). Larvae were categorized into four conditions: (i) dead, (ii) swimming, (iii) metamorphosed and floating in the water, i.e., premature metamorphosis (*sensu* Edmunds et al., 2001), and (iv) metamorphosed and firmly attached to CCA or dish, i.e., successful settlement.

**2.5 ~~Chlorophyll fluorescence and bleaching~~Recruit experiment**

To test the effects of temperature treatments on the photophysiology, growth and survival of recruits, a second batch of larvae were transferred to 10-cm-diameter petri dishes which were left floating in a flow-through tank at ambient temperature. Twenty hours later, 4 dishes with a total of 30–35 newly settled recruits were assigned to each treatment tank and placed at the bottom of treatment tanks. Only recruits that settled

individually and at least 1 cm apart from others were use for the experiment in order to avoid possible contact between recruits during growth. Dishes were rotated daily to avoid the potential positional effects within each tank.

Twenty 3-day-old recruits were randomly selected and marked in each treatment. Diving-pulse-amplitude modulation (PAM) fluorometry (Walz, Germany) was used to measure the maximum quantum yield of PSII ($F_v/F_m$), a proxy for potential photochemical efficiency of symbionts (Genty et al., 1989). Measurements were conducted at 05:30 on four consecutive days to allow enough time for dark adaption. Both the measuring light and gain of PAM settings were adjusted to "7" to give optimal fluorescence signals.

To better assess the photo-physiological performance of symbionts, effective quantum yield ($\Delta F/F_m'$) was also measured for 15 recruits from each treatment four times on the last day of the experiment (08:00, 11:00, 14:00, 17:00). The maximum excitation pressure over PSII ($Q_m$) was calculated using the equation: $Q_m = 1 - [(\Delta F/F_m'$ at $14:00)/(F_v/F_m)]$ (Iglesias-Prieto et al., 2004).

Bleaching response was assessed photographically following Siebeck et al. (2006) with some modifications. At the end of the experiment, recruits were photographed with a digital camera under the dissecting microscope and identical illumination (35 $\mu$mol photons m$^{-2}$ s$^{-1}$). The camera was set on manual mode with constant ISO settings (12800). Saturation of each coral picture, a good proxy for symbiont or chlorophyll density during bleaching, was measured by taking the average value of 30 randomly placed quadrats (100$\times$100 pixels each) on each coral picture using

Photoshop's histogram function (Siebeck et al., 2006). The total chlorophyll/symbiont content of each recruit was determined by multiplying the mean saturation by surface area (as measured ~~in Section 2.6~~ below) to account for differences in the size of recruits. Bleaching response was quantified as the reduction in chlorophyll/symbiont content of each recruit relative to the recruit yielding the maximum value.

## 2.6 Post-settlement survival and growth

Recruits were checked daily under a dissecting microscope throughout the experiment and scored as alive or dead based on the presence of polyp tissue. At each census, the number of living recruits was recorded for each treatment. Digital images of recruits with scale calibration were also analyzed for lateral growth using ImageJ software (National Institutes of Health). The number of polyps for each recruit was counted visually. Juvenile growth was estimated as the rates of change in planar area and number of new polyps over time (Dufault et al., 2012; Jiang et al., 2015).

Calcification was calculated as the dry skeletal weight deposited per day (Dufault et al., 2012). Tissue of recruits was removed with a water-pick at the end of the experiment. Skeletons were weighed individually using an ultra-microbalance at an accuracy of $\pm 1$ μg. Furthermore, the temperature coefficient ($Q_{10}$), which is widely used to express the sensitivity of metabolism, development and growth to temperature changes (Hochachka and Somero, 2002; Howe and Marshall, 2001; Rivest and Hofmann, 2014), was calculated using the equation: $Q_{10} = (R2/R1) \wedge (10/(T2-T1))$,

where R is the growth rate at temperature T2 or T1. $Q_{10}$ values of enzyme-catalyzed
reactions often double ~~for the~~with a 10 ℃ increase in temperature.

## 2.~~7~~6 Data analyses

Data were tested for homogeneity of variances, using Cochran's test, and normality
was assessed using Q−Q plots. Percent data in settlement assays and budding rates
were square root transformed to meet the requirements of homogeneity of variances.
Larval settlement, $Q_m$ and growth parameters were compared among treatments using
two-way analyses of variances (ANOVAs) with mean temperature and temperature
variability as fixed factors, each with two levels (29 and 31 ℃; constant and
fluctuating regimes). When main effects were significant ($P < 0.05$), planned multiple
comparisons were conducted using Fisher's LSD tests, which are more powerful than
the original ANOVA (Day and Quinn, 1989; Lesser, 2010). Recruits were divided
into 3 categories according to the number of polyps: 1-polyp, (2-4)-polyp and
(5-6)-polyp. A Chi-square test was used to compare the differences in bud formation
among treatments. Survivorship of coral recruits was analyzed using a Kaplan-Meier
(KM) log-rank analysis. Two-way ANOVAs with repeated measures were used to test
for the effects of temperature treatments on $F_v/F_m$ or $\Delta F/F_m{}'$ over sampling time points.
All statistical analyses were performed with STATISTICA version 12.0 (Statsoft).

## 3 Results

**3.1 Larval settlement**

During the settlement assays, lLarval mortality was only observed in the constant 31 ℃ treatment during the settlement assay (Fig. 2a). In all treatments, between 35 and 60% of larvae metamorphosed whilst in a free-floating polyp state (Fig. S2), and between 2.5 and 15% were swimming actively (Fig. 2b). Although the differences in these percentages among treatments were not significant (Table S1), there were more metamorphosed and floating larvae in the constant 31 ℃ treatment than in other treatments. Settlement was significantly affected by elevated temperature and marginally affected by the interaction between temperature level and regime (Table S1-). Specifically, percent settlement was similar between the two temperature regimes at 29 ℃, but differed between the constant and fluctuating treatments at 31 ℃. The settlement rate at fluctuating 31 ℃ was comparable to that in the control treatment and significantly higher than that in the constant 31 ℃ treatment (Fig. 2c, Table S2-).

**3.2 Chlorophyll fluorescence and bleachingPhoto-physiology, growth and survival of recruits**

A significant interaction between time, temperature level and regime was observed for maximum quantum yield $F_v/F_m$ (Table S3-, Fig. 3a). Separation of the results by time showed that $F_v/F_m$ was consistently lower at higher temperatures, but the effect size was small, only amounting to a 3 % decrease (Table S4-). There was also a significant

interaction between time, temperature level and temperature regime for effective

quantum yield $\Delta F/F_m'$ (~~~~Table S3~~.~~). Further separate analyses revealed that both

temperature increase and fluctuations had strong effects except at 08:00 (Table S4~~.~~),

with lower $\Delta F/F_m'$ at elevated temperature and higher $\Delta F/F_m'$ under fluctuating

conditions (Fig. 3b).

$Q_m$, the maximum excitation pressure, was not influenced by elevated temperature

(Table S5~~.~~). However, it was considerably reduced under fluctuating regimes (Fig. 3c,

Table S5~~.~~). Recruits at 31 ℃ exhibited a paler appearance than those at 29 ℃, as

evidenced by the reduction in saturation and increase in brightness (Fig. S3). However,

bleaching index which accounts for differences in recruit size, was unaffected by

temperature level, regime,  or their interaction (Fig. 3d, Table S5~~.~~)

### ~~3.3 Growth, survival and Q~~$_{10}$

The budding state of recruits differed significantly among treatments (Chi-square

test, $\chi^2 = 19.4$, $df = 6$, $P = 0.004$). Seven days after settlement, approximately 70% of

recruits at 31 ℃ produced at least one bud, compared to less than 50% of recruits at

29 ℃ (Fig. 4a). Budding rates at 31 ℃ were more than twice those at 29 ℃ (Fig. 4b,

Table ~~S6~~S5~~.~~). No significant differences between the constant and fluctuating regimes

were observed at either temperature ~~(Table S6.)~~.

Lateral growth rates increased significantly with elevated temperature~~-~~, but were

not affected by temperature fluctuations (Fig. 4c, Table ~~S6~~S5~~.~~). The skeletal weight

deposited each day was 56% higher at 31 ℃ than at 29 ℃ (Table ~~S8~~S5~~.~~). The effects

of temperature fluctuations on calcification ~~were dependent~~depended on the mean

temperature (Fig. 4d), ~~although~~even though the interaction between temperature level

and regime was not statistically significant (Table ~~S6~~S5~~.~~)~~. At:~~ at 29 ℃, the fluctuating

~~regime~~treatment had no discernible effect on calcification, while in the fluctuating

regime with a mean temperature of~~at~~ 31 ℃ ~~it caused~~a significant reduction (20%) in

calcification was observed when compared to the constant 31 ℃ regime (Table

~~S7~~S6~~.~~).

Survival of recruits remained >86% in all treatments after 7 days, with the highest

and lowest values at 31 ℃ (97%) and 29 ℃ (86%), respectively. Survivorship did not

vary significantly across treatments ($\chi^2$ = 4.49, $df$ = 3, $P$ = 0.21, Fig. 5), although it

was 6–13% higher at elevated temperature. For juvenile *P. damicornis*, lateral growth,

budding and calcification increased by 1.19-, 1.91- and 1.68-fold respectively

between 29 and 31 ℃, yielding a $Q_{10}$ of 2.6, 36.8 and 17.8.

**4 Discussion**

**4.1 Larval settlement under elevated and fluctuating temperatures**

The pronounced declines in successful settlement at constant 31 ℃ were consistent

with previous findings that reported the effects of thermal stress (>30 ℃) on coral

larval settlement (Humanes et al., 2016; Randall and Szmant, 2009). Interestingly,

transient exposure to 33 ℃ in variable conditions did not produce the same negative

effect on larval settlement as constant exposure to 31 ℃; on the contrary, coral larvae

experiencing diurnal shifts between 30 and 33 ℃ settled at a similar rate to those in
the control. During daytime exposure to elevated and stressful temperatures, coral
larvae may not initiate metamorphosis and settlement because larvae undergoing this
complex stage are particularly susceptible to thermal perturbations (Randall and
Szmant, 2009), but settlement may proceed as temperature descends to a more
tolerable level at night (30 ℃ in this study). It is likely that the fluctuating
temperature conditions could provide some respite for coral larvae, thereby favoring
settlement at elevated and fluctuating temperature conditions. More precise
assessment of settlement timing was not possible without disturbing larvae, given the
use of small petri dishes. Future studies are needed to regularly observe and establish
the dynamics of larval behavior under fluctuating temperatures to confirm this
hypothesis.
Another possible cause for the higher settlement of larvae in the fluctuating 31 ℃
treatment may be the brief exposure to extreme temperatures around noon. Previous
studies have demonstrated that short-term exposure (minutes to hours) of coral larvae
to extremely high temperatures (33-37 ℃) significantly enhanced the subsequent
settlement at lower temperature, suggesting a strong latent effect (Coles, 1985;
Nozawa and Harrison, 2007). Therefore, the 2-hour incubation at 33 ℃ during the
daytime may have exerted a latent and stimulatory effect on settlement at night when
the temperature was lower.
Metamorphosed and floating larvae, previously noted in corals (Edmunds et al.,
2001; Vermeij, 2009; Mizrahi et al., 2014; Richmond, 1985), were more frequent at
elevated temperatures. One possible explanation is that premature metamorphosis in
coral larvae is a spontaneous response to increased temperatures (Edmunds et al.,
2001). The floating polyps, as a result of pelagic metamorphosis, have been shown to
have extended longevity, possibly because they can obtain energy from
photosynthesis by maternally derived symbionts and heterotrophic feeding using
tentacles (Mizrahi et al., 2014; Richmond, 1985). Thus, plasticity of metamorphosis
during the dispersive phase could be a strategy for coping with environmental stress in
coral larvae, although it remains to be determined whether these floating polyps are
capable of settling and contributing to recruitment in natural conditions.

**4.2 Symbiont responses to elevated and fluctuating temperatures**
The reduction in $F_v/F_m$ at 31 ℃ does not indicate severe damage to the photosynthetic
apparatus or chronic photoinhibition, as the values were still within the healthy range
(Hill and Ralph, 2005). The fluctuating regime had positive effects on $\Delta F/F_m^{'}$,
suggesting a greater light use efficiency to drive photochemical processes. $Q_m$, an
indicator of the excitation pressure over PSII, was reduced in fluctuating treatments,
reflecting a stronger competitiveness of photochemical process for reaction centers
over nonphotochemical quenching (Iglesias-Prieto et al., 2004). The higher $\Delta F/F_m^{'}$
and lowered $Q_m$ under fluctuating conditions suggest that the diel temperature
oscillations could relieve heat stress on corals and corroborate previous findings that
temperature fluctuations are favorable to the photo-physiology of corals (Mayfield et
al., 2012; Putnam et al., 2010). The positive effect of exposure to fluctuating
temperatures on these photo-physiological metrics may be associated with the cooling
overnight and upregulation of the genes related to photosynthesis (Mayfield et al.,

2012).

In contrast to the aforementioned studies, Putnam and Edmunds (2008) found that

when incubated at fluctuating temperatures (26–32 ℃), $F_v/F_m$ of *P. meandrina* and
*Porites rus* nubbins were depressed by ~20% compared to those maintained at a
constant temperature of 28 ℃. These contrasting results may be due to
methodological differences. Our study and Mayfield et al. (2012) mimicked natural
temperature fluctuations by progressively modulating temperatures over time,
whereas Putnam and Edmunds (2008) directly transferred corals from low to high
temperature in the morning and vice versa at night. This approach could cause instant
heat-shock and prolonged exposure to extreme temperatures, thereby exaggerating the
stressful effects of diurnal thermal fluctuations.

Although juvenile *P. damicornis* at 31 ℃ exhibited apparent paling appearance

compared to those in 29 ℃, loss of symbionts and bleaching were not indicated, as
the faster lateral growth at 31 ℃ suggests that the paling is instead the result of
pigment dilution due to a larger surface area. This outcome contrasts with previous
work showing the sensitivity of endosymbionts within coral recruits to elevated
temperatures (Anlauf et al., 2011; Inoue et al., 2012). The lack of bleaching response
to elevated temperatures in the current study may be linked to the symbiont type. *P.*
*damicornis* predominantly harbored *Symbiodinium* clade D in Luhuitou (Zhou, 2011),
which has been found to be particularly thermally tolerant. In addition, the difference
in treatment duration could also partially explain these contrasting sensitivities. Albeit
ecologically relevant, the exposure duration in this study was much shorter than
previous studies (Anlauf et al., 2011; Inoue et al., 2012), therefore resulting in less
cumulative stress. It is possible that a longer exposure time may cause similar
bleaching responses to those found by other studies.
Further, daytime exposure to high temperatures in fluctuating treatments did not
induce significant symbiont loss in juvenile *P. damicornis*. This observation is in stark
contrast to the observations of Putnam and Edmunds (2011) on adult corals. That
study found that ephemeral exposure to 30 ℃ at noon in fluctuating conditions (26–
30 ℃) elicited a 45% reduction in symbiont density of adult *P. meandrina* compared
to corals at the steady 28 ℃ treatment, a larger effect than that was elicited by
continuous exposure to 30 ℃ (36%). The flat structure of juvenile corals has been
suggested to provide a higher mass transfer capacity to remove reactive oxygen
species than the branching and three-dimensional adults (Loya et al., 2001). Hence,
the discrepancy between our results and that of Putnam and Edmunds (2011) may, at
least partially, be attributed to the morphology-specific difference in thermal tolerance
of juvenile and adult corals.

**4.3 Accelerated early development at elevated temperature**
Early development of juvenile *P. damicornis*, including budding, lateral growth and
calcification, was accelerated at 31°C, which is 2°C above the local long-term
summer mean and 1°C above the local bleaching threshold (Li et al., 2012). Growth
stimulation by temperature increase also occurred in a pilot study which showed that
lateral growth and budding of *P. damicornis* after two weeks at 31°C were 10% and
41% higher respectively than that of those at 29°C (Fig. S4). Moreover, recruits with
increased growth rates at elevated temperatures showed higher survivorship,
consistent with previous field observations that survival in early stages of reef corals
was strongly dependent on colony size and growth rates (Babcock and Mundy, 1996;
Hughes and Jackson, 1985). In contrast to our study with a tropical coral, a previous
study reported that calcification of symbiotic polyps of *Acropora digitifera* in
subtropical Okinawa was highest at 29°C (2°C above the local summer mean), and
was reduced at 31°C (Inoue et al., 2012).
It has been widely accepted that warming is likely to be more deleterious to early
stages of tropical corals than subtropical species (Woolsey et al., 2014). Clearly,
thermal tolerance of corals ~~is relative to~~depends on the ambient temperature at a
particular location. Given the large seasonal temperature fluctuations and ranges in
our study site (Fig. S1), it is not surprising that *P. damicornis* grew faster at 31°C.
The positive effects of the 2°C temperature increase on the early development of *P.*
*damicornis* suggest that tropical corals dwelling in thermally dynamic habitats may
also have the capacity to modify their thermal limits, thereby enhancing physiological
performance and tolerance under increasing temperatures (Clausen and Roth, 1975;
Dandan et al., 2015; Schoepf et al., 2015).
There are two possible explanations for the increases in growth and development at
elevated temperature in our study. Firstly, paling of recruits at elevated temperatures
as a result of pigment dilution will enhance their internal light fields, which could
bring about 2- to 3-fold increase in symbiont specific productivity (Wangpraseurt et
al., 2017), and in turn support skeletal growth and asexual budding. Secondly, since
coral calcification is positively correlated with carbon translocation between
*Symbiodinium* and the host (Tremblay et al., 2016), the elevated calcification and
growth at 31 ℃ indicates more efficient nutritional exchange, sustaining the
metabolic expenditure of faster development. This interpretation is further supported
by the excessive deviation of $Q_{10}$ from the kinetic expectations (2–3): this signifies a
strong amplifying effect through changes in fundamental biochemical systems along
with the acceleration of functional enzyme activities at increased temperatures
(Hochachka and Somero, 2002).

**4.4 Differing effects of temperature fluctuations on growth**

The growth-related processes, including budding, lateral growth and calcification
differ in their responses to temperature fluctuations, with calcification being more
responsive. The lack of statistically significant effects of temperature fluctuations on
budding and lateral growth suggests that either these processes were not affected by
fluctuating temperatures, or the length of exposure to the peak temperatures may be
not long enough to trigger a detectable effect (Lucas and Costlow, 1979).
The impact of fluctuating temperatures on calcification was different at ambient
and elevated temperatures: the fluctuating treatment did not affect calcification at
29 ℃, but resulted in a significant decline at 31 ℃. In comparison, prior studies with
corals did not find that temperature fluctuations influenced skeletal growth (Mayfield
et al., 2012; Putnam and Edmunds, 2011). It is likely that the impact of temperature
fluctuations depends critically on whether the temperature range encompasses the
maximum thermal limits of the species (Vasseur et al., 2014).
The relationship between skeletal growth in corals and temperature is non-linear
and characterized by a parabola ~~whose apogee indicated~~with an optimum and
threshold, beyond which the stimulatory impact of temperature will be reversed
(Buddemeier et al., 2008; Castillo et al., 2014; Inoue et al., 2012; Wúm et al., 2007).
Although the optimal temperature for calcification by *P. damicornis* recruits remains
unknown, it is possible that the recruits exposed to the fluctuating 31 ℃ treatment
calcified at a slower rate when the temperature was below 31 ℃ compared to those in
the constant 31 ℃. However, given the well-established temperature performance
curve for coral calcification (Buddemeier et al., 2008; Wúm et al., 2007), daytime
exposure to temperatures above 32 ℃ would have severely impaired the calcification
process, thus leading to an overall decrease in calcification. At least two hypotheses
from the literature can help explain this inhibitory effect. First, during the warmest
~~hottest~~ part of a daily temperature cycle, metabolic rates will usually be depressed to
improve energy conservation (Marshall and McQuaid, 2010; Putnam and Edmunds,
2008; Sastry, 1979). Depression in metabolism and ATP production in this specific
"quiescent" period may impose constraints on daytime calcification, as calcification is
energetically costly, consuming up to 30% of the coral's energy budget (Allemand et
al., 2011). An alternative and nonexclusive explanation is that daytime exposure to
extreme temperature could disturb the function and/or synthesis of skeletal organic
matrix (OM) within the calcifying medium. The OM has critical roles in calcification
such as calcium binding, providing carbonic anhydrase and the template for crystal
nucleation (Allemand et al., 2011). Daytime temperatures of 33 ℃ may disrupt the
function of carbonic anhydrases (Graham et al., 2015), thereby severely inhibiting the
conversion of respired $CO_2$ to bicarbonate for subsequent use in calcification.
Further, since the OM itself is also incorporated into the skeleton, the rate of OM
synthesis is a limiting factor for calcification (Puverel et al., 2005; Allemand et al.,
2011). Extreme temperatures may impede the production of OM as it is highly
sensitive and vulnerable to short-term thermal stress (Desalvo et al., 2010; Desalvo et
al., 2008; Maor-Landaw et al., 2014). Although the exact mechanism has not yet been
fully resolved, our study provides evidence that daytime exposure to extreme
temperature in variable thermal conditions adversely affects calcification, and
dampens the stimulation of skeletal growth in *P. damicornis* at elevated temperature.

**5 Conclusions and implications**
This study was the first to examine the effects of both increased temperature and daily
temperature variability on the early stages of a reef coral. We found that realistic

diurnal temperature fluctuations considerably tempered thermal stress on larval

settlement, and had varied effects on the physiology and early development of *P.*

*damicornis*. Diel oscillations in temperature did not induce bleaching but relieved heat

stress on photo-physiology. Further, temperature fluctuations had no obvious effects

on budding and lateral growth, although two hours' exposure to 33 ℃ during the

daytime apparently caused a reduction in calcification compared to constant exposure

to 31 ℃. Results reported here emphasize the distinction between the effects of

constant and fluctuating temperatures, both for different mean temperatures and on

two successive life stages, and highlight the importance of incorporating diurnal

fluctuations into research on the influence of ocean warming on coral biology.

The results of this study suggested that coral larvae subjected to diurnal

temperature variations, especially at increased temperature, exhibit better settlement

competence than those subjected to static thermal treatment. The fluctuating

temperatures were favorable to the photo-physiology of endosymbionts and only had

minor effects on post-settlement development of coral recruits. Therefore, ~~for~~ corals

in highly fluctuating environments~~, they~~ may have the potential to tolerate and

acclimate to the changing seawater temperatures. These findings may also provide

clues as to how diverse coral communities can persist and thrive in some thermally

variable conditions (Craig et al., 2001; Richards et al., 2015). It is important to note

that this study was technically limited to only one fluctuating amplitude, and the

extent of thermal variance has as much of an impact on fitness as the changes in mean

temperature (Vasseur et al., 2014). Given that there is currently still no consensus on

the future temperature variability (Burroughs, 2007), it will be critical to study the
impact of a broad range of thermal variations which corals may fare in a warming
ocean.

**Data availability**
The data associated with the present study is available from the corresponding author
upon request.

**Author contributions**
L. J. and H. H. conceived and designed the experiments; L. J., Y. F. S., and Y. Y. Z.
performed the experiments; X. B. L., L. J. M., J. S. L., X. M. L., G.W. Z., S. L., and P.
Y. Q. contributed analysis and materials. L. J wrote the manuscript with comments
from all co-authors.

**Competing interests**
The authors declare that they have no conflict of interest.

**Acknowledgements**
This work was funded by National Natural Science Foundation of China (U1301232,
41206140, 41306144 and 41476134)~~, Science and Technology Service Network~~
~~Initiative (KFJ-EW-STS-123)~~ and Science and Technology Planning Project of
Guangdong Province, China (2014B030301064). L. J. McCook was supported by a
President's International Visiting Expert Professorial Fellowship from the Chinese
Academy of Sciences (2016VEA025). We are grateful to various reviewers for their
valuable comments and suggestions which improved this manuscript significantly.

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

**Figures and captions**
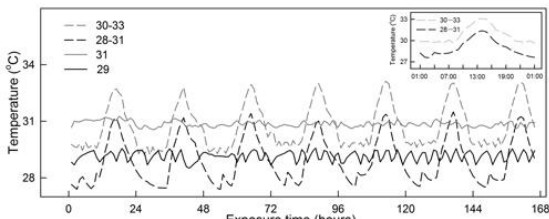
**Fig 1.** Temperature profiles for each treatment throughout the experiment. The inset
shows the one-day temperature trajectory in the two oscillating treatments. Time
course in fluctuating treatments was: 10 h at minimum temperature; 5 h of upward
ramping; 2 h at maximum temperature; 7 h of downward ramping (passive).

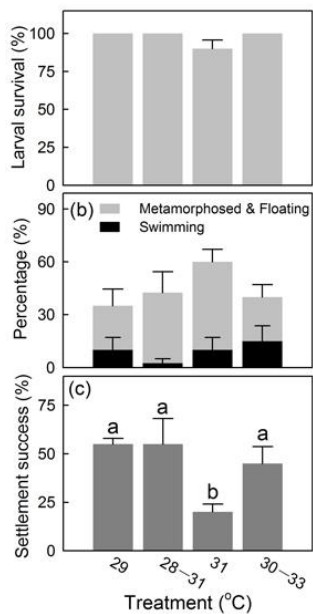
**Fig 2.** Percentage of *P. damicornis* larvae that (a) survived, (b) metamorphosed while
floating and remained pear-shaped, and (c) successfully settled after 24 h exposure to
temperature treatments. Error bars represent 1SEM (n = 4). Different letters denote
significant differences between treatments.

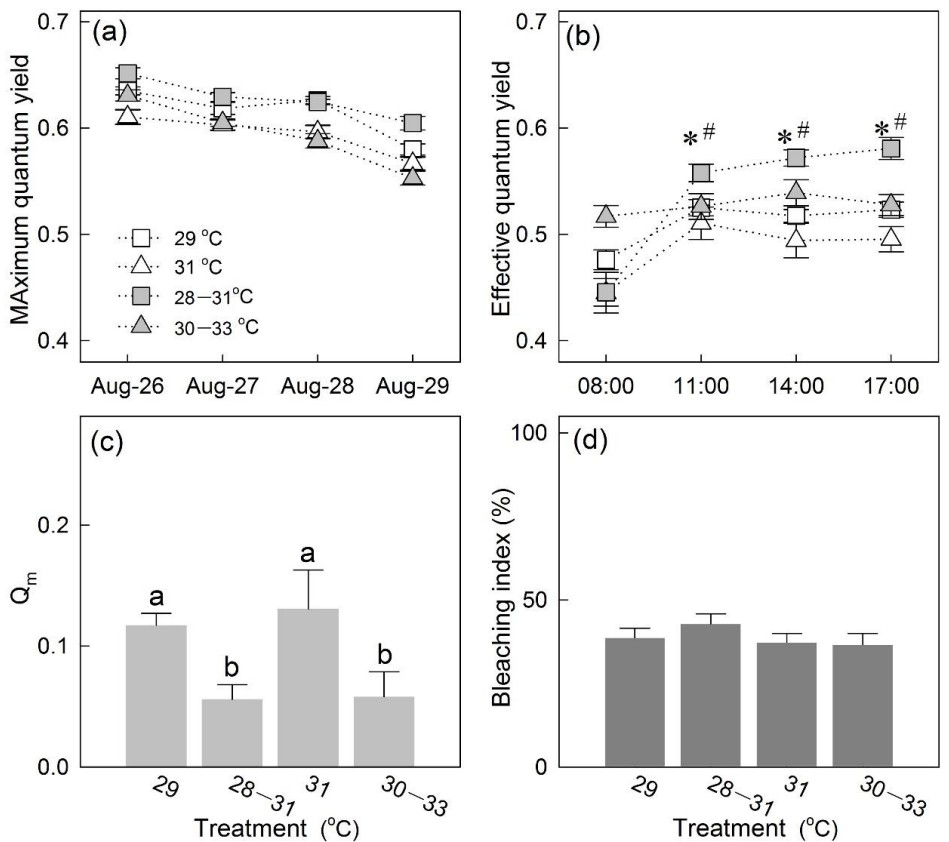

**Fig 3.** Photo-physiology and bleaching of *P. damicornis* recruits under constant and fluctuating conditions of two temperatures (29 and 31 ℃). (a) $F_v/F_m$ over four consecutive days, (b) $\Delta F/F_m'$ throughout the last day of the experiment, (c) $Q_m$ and (d) bleaching rates. Error bars represent 1SEM (n = 20 for $F_v/F_m$; n = 15 for $\Delta F/F_m'$ and $Q_m$; n = 25−33 for bleaching index). Asterisks and hashes indicate significant effects of temperature increase and fluctuations at a specific time, respectively. Different letters represent significant differences between treatments.

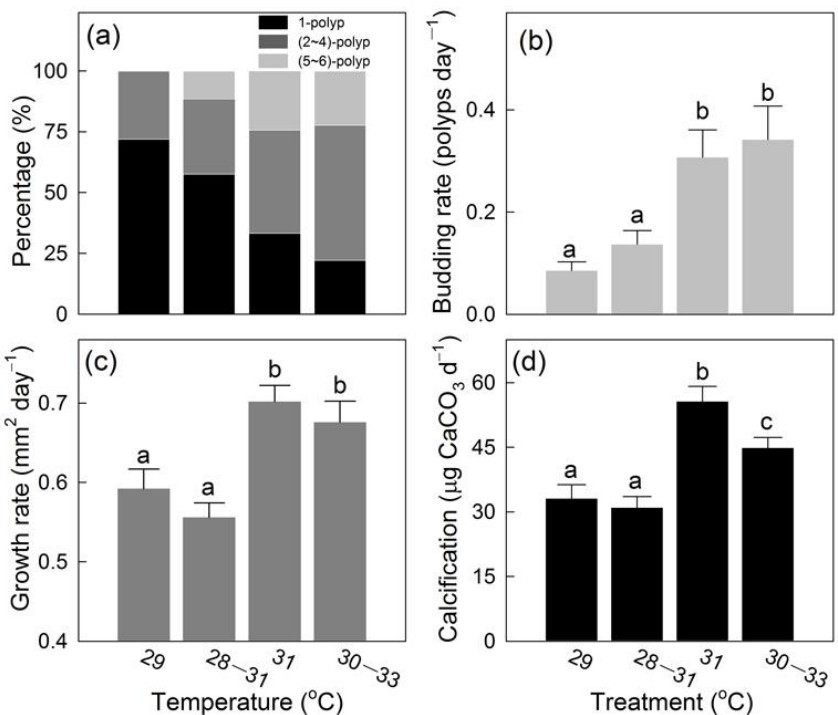

**Fig 4.** (a) Budding state, (b) polyp formation rate, (c) lateral growth and (d) calcification of *P. damicornis* recruits under constant and fluctuating conditions of two temperatures (29 and 31 ℃). Error bars represent 1SEM (n = 25–33). Different letters denote significant differences between treatments.

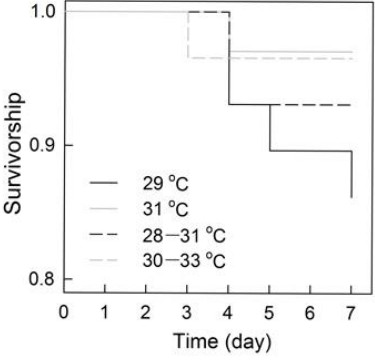

**Fig 5.** Survivorship of *P. damicornis* recruits estimated using Kplan-Meier analysis in

each treatment over the 7-day experiment. The numbers of recruits at the start of the

experiment in each treatment are 30 for the treatments 29 ℃, 30–33 ℃ and 28–31 ℃,

and 35 for 31 ℃