# Peer review of "the reef coral Pocillopora damicornis"

_Biogeosciences, 2017_

## Referee Comment (RC1) · D. Barshis (Referee) · 8 Jun 2017

General comments: The authors present a comprehensive assessment of the role of diurnally fluctuating temperatures on growth, settlement, and bleaching response of larvae from the coral Pocillopora damicornis. The study is quite sound and represents an important contribution to the field. Most coral thermal stress studies use static temperature exposures, hence a movement in the field to more realistic natural thermal profiles is desperately needed. Yet we still lack a fundamental understanding of the different responses of corals to static or variable temperatures in the same study. This research begins to fill in that gap and the manuscript is technically sound and

well-presented. There are a few minor comments that should be addressed prior to publication as well as an additional reference that should be integrated into the discussion on growth (see line-by-line comments below). Also, while the writing is generally sound, there are a few instances of misuse of the word "the" and singular/plural errors that may be resolved by additional editing of the language. All in all, I think this is a sound paper that makes an important and needed contribution to the literature.

Specific line-by-line comments: Line 197. Siebeck found brightness and saturation to be indicative of bleaching, why was only saturation used?

Section 2.7 Please specify the software used for statistical tests and copies of code (as supplementary information) if possible.

Line 250-252. Confusing wording. Please clarify that both the elevated 31 °C stable and 30-33 °C fluctuating treatments induced bleaching while the control and 28-31 °C fluctuating treatments did not.

Lines 327-332. Would add discussion of the increased growth and survival in the higher temps. They may have decreased in color saturation but were not "stressed" according to the other metrics. There could also be a confound wherein a faster growing colony might pale simply because it's growing faster than the Symbiodinium are dividing so its not losing cells, just diluting pigment. The photographic technique here does not allow for analysis of cell loss and it's unclear over how much area saturation was measured (i.e. how many pixels) and whether it was normalized to surface area or polyp number to account for size differences.

Section 4.4 Please see Buddemeier et al 2008 A modeling tool to evaluate regional coral reef responses to changes in climate and ocean chemistry. Limnology and Oceanography Methods. Particularly their meta-analysis in Figure 2. An alternative explanation may simply be a decreasing slope of the temperature x calcification relationship at higher temperatures as you approach the optimum (Buddemeier Fig. 2), wherein the corals are not calcifying linearly within the temperature fluctuation (i.e. at

temperatures above the mean they're not growing much faster and they are growing slower at temperatures below the mean thus resulting in overall decreased calcification in comparison to 31 stable).

―――――――――――――――――――――

---

## Referee Comment (RC2) · E. Rivest (Referee) · 23 Jun 2017

General comments

In their manuscript titled "Impact of diurnal temperature fluctuations on larval settlement and growth of the reef coral Pocillopora damicornis," the authors present research on an exciting and timely topic – the effect of temperature variability on thermotolerance of two life history stages of a common reef-building coral. The topic is within the scope of the journal and the focus on effects of environmental variability is still novel within the coral field. Unfortunately, I find that this paper is not suitable for publication in its present form. There are several general ways in which this manuscript can be improved.

[Figure]

1. The Introduction should include a description of the study species and of their reproduction (brooding) and the fact that the larvae contain symbionts upon release. These are critical pieces of information that the general readership of Biogeosciences will likely not know and are important for properly interpreting the results.

2. The Methods needs a much better overall description of the experimental design. It is difficult to tell if the spat were from the same or separate trials. Furthermore, the experimental design is flawed because it does not include replication of the treatments and the culturing techniques are not shown to avoid imposing artifacts on the responses of the corals.

3. The statistical tests and results need to be fully described. Posthoc analyses are not described. Table(s) with full results of all statistical models should be included, including results of posthoc analyses.

4. More synthesis and integrative discussion is needed across all the responses measured to inform a broader picture of the implications for the ecology of this coral. The authors need to place their results in the broader context of biogeosciences and coral reef ecology.

Specific comments

Introduction

L58-59 – "sea surface temperature have increased on average by 0.7deg C"...since what date? A frame of reference is needed here.

L65-70 – it would be good to cite studies that have quantitatively analyzed temperature variability for coral reefs here like Rivest and Gouhier, 2015 and Guadayol et al. 2014

L77-79 – actually, there are a handful of studies (at least 7) that have looked at the effects of temperature variability. I do see that the authors have described the results of a few of these studies in the next paragraph, but they should rephrase this sentence to better define the knowledge gap that their study aims to fill.

L83 – "more suited" is vague and confusing. Please be more specific here.

L84 – "deleterious effects" of what? Diel temperature oscillations?

L86 – "under diel temperature oscillations" compared to what?

L90-93 – this statement needs references.

Methods

L126 – the date of collection of adult corals and the holding conditions of the corals prior to larval release need to be included. The temperature of the water at which the larvae were released should be included.

L129 – "the recruit experiment" – is this the settlement or post-settlement experiment? This should be more clearly defined using a phrase like "to test the effects of xx on yy, larvae were transferred". This is confusing to the reader because the authors have not defined what settlers or recruits are. Remember – the audience is general and interdisciplinary. Or perhaps it would be more clear to describe more generally that the larvae and settlers are being tested in completely separate experiments?

L130 – were the dishes covered? Did the authors account for/measure effects of evaporation on salinity? Did the authors measure the temperature in the floating dishes during this time? Was there selection that could have influenced the performance of the spat? Again, "spat" is another new synonym used. Please choose one term for the juvenile corals, define it clearly for the reader, and use it consistently throughout the text.

L135 – "ambient temperature" where? At the collection site of the adult corals?

L153-155 – these are results and should be moved to that section.

L155 – how was salinity checked?

L159-162 – these are results and should be moved to that section.

L162 – it is a significant limitation that the experiment has no true replication. I understand and empathize with the frustrations of facility and logistical constraints but more justification is needed for the validity of the results. Could the authors repeat the experiment to replicate the results in place of replication during the experiment?

L166 – the title "Settlement assay" makes me think that the authors are going to be testing effects on settlement and is confusing with "preparation of spat" in the title of the last section. Please revise.

L168 – is this species of CCA a natural settlement substrate for this species in your location? Please provide additional details here.

L170 – did the dishes have lids? Were they sealed in the treatment tank ("submerged")? What was the depth of the water in the dishes? It seems like a very high spat density in a small volume of water. Please provide justification that these are natural and representative settlement conditions for this species.

L180 – where did these spat come from? Were they from the "settlement assay" or from "preparation of spat"? Were they kept in the four treatments during this time? I can't interpret the results of these tests without knowing these important details.

L194-195 – describe the settings for photography and illumination to allow others to replicate your measurements.

L198 – the statistical comparison needs to be described here. What were the controls? Was the bleaching index assessed as relative to corals in the control treatment or was it a comparison of absolute values?

L201 – which recruits? The ones assessed for bleaching? Different ones?

L213 – details of posthoc analyses need to be included.

Results

L229-230 – is this 'normal' settlement behavior for this species? Could it be an artifact

of the 'unnatural' settlement conditions?

L231-235 – since the results were not significant, there are no "distinct" differences. If the interaction is not significant, how can there be significant groupings stated on the figure (2c)?

L237 – "greatly alleviated" is an interpretation and does not belong in the Results section. The phrase "in contrast" is inappropriate here because settlement success was not statistically distinct with that under the fluctuating and constant regimes at 29degC.

L241 – what were the separate analyses?

L255 – replace "strongly" with "significantly." Also, the Chi-square test was not listed in the Results section. Please include.

L264-267 – again how can the authors claim this if the model was not statistically significant?

L270 – survival of what?

L275 – this is the first time Q10 is mentioned. This needs to be included in the methods and defined carefully for the broad readership. Why was Q10 calculated for these results and not the others?

Discussion

L279 –Based on my interpretation of the data, it was only lower at constant elevated temperatures.

L282 – "hardly impaired" – too qualitative

L283 – I am having difficulty with the phrase "greatly attenuated the thermal stress on settlement" throughout the manuscript (alleviated, mitigated, tempered...). Because of the lack of replication, it is hard to attribute the responses to thermal stress and constant vs. variable conditions. I think it would be better to say something like "did

not produce the same negative response to high temperature as under exposure to constant high temperature." Based on the experimental design, it is impossible to know whether the corals simply experienced less thermal stress overall because they spent some time at temperatures less than 31degC each day or if they responded differently to the high temperature. These mechanistic possibilities should be discussed and phrasing should be more careful.

L288 – I don't think the authors can say that fluctuating conditions favor settlement because the 29degC constant and fluctuating conditions produced statistically similar settlement rates. Furthermore, when did settlement happen? Did it happen during the daytime when temperatures were higher, or during the nighttime when temperatures were lower? These details could be important for appropriate interpretation of the results.

L298-301 – what about the desperate larval hypothesis?

L327 – both constant and fluctuating T treatments

L340-342 – this sentence needs to be better integrated with the paragraph

L344 – this section does not mesh well with the rest of the Discussion

L407-410 – but calcification rates increased under the high temperature treatments. . .?

L429 – but it was still elevated compared to the 29degC treatments...

Figure S1. Panels a and b are not very relevant displays of temperature information for useful interpretation of the experimental design. A plot showing average seasonal daily temperature variability would be more useful. Plot d needs to have an x-axis label.

Technical comments

L116 – Doesn't the dataset go to 2016, not 2015?

L123 – Should Fig. S1d be cited here instead of S1c?

There are consistent errors in grammar and word choice throughout the manuscript. While it does not impede the reader from understanding the scientific content, I advise the authors to carefully copy edit the entire text.

---

## Author Comment (AC1) · 28 Jul 2017

**Authors' response to reviewers' comments on the manuscript bg-2017-120 "Impact of diurnal temperature fluctuations on larval settlement and growth of the reef coral *Pocillopora damicornis*" by Lei Jiang et al.**

**To the Editor**

Dear Dr. Christine Klaas,

We would express our sincerest gratitude for your help to correct some errors in the early version of this manuscript, and all the time and efforts it took to develop this manuscript and the review process. Furthermore, we appreciate the constructive comment from the two reviewers. We have carefully considered and incorporated the comments and suggestions from both reviewers, and the point-by-point responses are as follows. We are looking forward to receiving your response soon.

Best wishes,
Lei Jiang on behalf of all authors, jianglei12@mails.ucas.ac.cn

**To Referee#1 Dr. D. Barshis**

[*General comments*] *The authors present a comprehensive assessment of the role of diurnally fluctuating temperatures on growth, settlement, and bleaching response of larvae from the coral* Pocillopora damicornis. *The study is quite sound and represents an important contribution to the field. Most coral thermal stress studies use static temperature exposures, hence a movement in the field to more realistic natural thermal profiles is desperately needed. Yet we still lack a fundamental understanding of the different responses of corals to static or variable temperatures in the same study. This research begins to fill in that gap and the manuscript is technically sound and well-presented. There are a few minor comments that should be addressed prior to publication as well as an additional reference that should be integrated into the discussion on growth (see line-by-line comments below). Also, while the writing is generally sound, there are a few instances of misuse of the word "the" and singular/plural errors that may be resolved by additional editing of the language. All in all, I think this is a sound paper that makes an important and needed contribution to the literature.*

[**Reply**] Thanks for the positive comments regarding our manuscript and other insightful and helpful suggestions. We will try our best to integrate all the constructive suggestions and further resolve the mistakes about the wording and singular/plural errors.

*Reply to specific line-by-line comments:*

[*Comment 1*] *Line 197. Siebeck found brightness and saturation to be indicative of bleaching, why was only saturation used?*

[**Reply**] Work by Siebeck *et al.*, 2006 suggested that for pictures of bleached *Pocillopora damicornis*, there were reduced saturation and elevated brightness values. Here, we measured the saturation and brightness values simultaneously and observed the reduction in saturation and increase in brightness (Fig S2). We only presented the saturation value to illustrate the paling of corals at elevated temperatures in the manuscript. We will further include the data on saturation and brightness in **Supplement (Fig. S2)**. Please refer to the Fig. S2 below for further details.

[Figure]

Fig. S2 Photographic metrics for *Pocillopora damicornis* recruits at different

temperature treatments.

[**Comment 2**] *Section 2.7 Please specify the software used for statistical tests and copies of code (as supplementary information) if possible.*

[**Reply**] All statistical analyses were performed with STATISTICA version 12.0 (Statsoft). This will be clarified in the text and **Supplement**.

[**Comment 3**] *Line 250-252. Confusing wording. Please clarify that both the elevated 31 °C stable and 30-33 °C fluctuating treatments induced bleaching while the control and 28-31 °C fluctuating treatments did not.*

[**Reply**] Revised as suggested. Now it reads "Compared to those in 29 °C, recruits at 31 °C showed a paler appearance as evidenced by the reduction in saturation and increase in brightness (Fig. S2). However, bleaching index which accounts for the difference in surface area of recruits, was unaffected by temperature level, regimes or their interaction (Fig. 3d)". Please see [**Reply**] to [**Comment 4**] below for explanations and details.

[**Comment 4**] *Lines 327-332. Would add discussion of the increased growth and survival in the higher temps. They may have decreased in color saturation but were not "stressed" according to the other metrics. There could also be a confound wherein a faster growing colony might pale simply because it's growing faster than the* Symbiodinium *are dividing so it's not losing cells, just diluting pigment. The photographic technique here does not allow for analysis of cell loss and it's unclear over how much area saturation was measured (i.e. how many pixels) and whether it was normalized to surface area or polyp number to account for size differences.*

[**Reply**] The discussion of increased growth and survival at higher temperatures will be added as follows, "Although extrinsic mortality risk was minimized in the present laboratory study, recruits with increased growth rates at elevated temperatures had higher survivorship. This finding was congruent with the paradigm that mortality in early stages of reef corals was strongly dependent on colony size and growth rates (Hughes & Jackson 1985; Babcock & Mundy 1996)".

After carefully examining our data, results totally supported the idea of the reviewer that coral recruits just became paling because of the faster growth. We are thankful to the reviewer for pointing out this puzzle and error. Generally, saturation and brightness of each recruit, were measured by taking the average value of 30 randomly placed quadrats (100×100 pixels each) on each coral picture using Photoshop's histogram function (Siebeck et al., 2006). The quantification of bleaching rates in coral recruits was quite different from that was employed for adult branches in Siebeck et al., 2006. For adult branches, the mean saturation values can be taken as the proxy for symbiont density, however, for the new recruits here, only the saturation cannot totally reflect the symbiont content. The bleaching index should consider the change in total content

rather than the mean density, because all recruits came from a single coral larva and recruits had significantly different surface area after exposure to different temperature conditions. Therefore, to account for the size difference between different treatments, the total chlorophyll/symbiont content of each recruit was determined by multiplying the mean saturation by surface area (as measured in Section 2.6). Bleaching response can be further quantified as the reduction in chlorophyll/symbiont content of each recruit relative to the one yielding the maximum value. Since we got similar results from both saturation and brightness measurements, we only presented that results calculated from saturation in Fig. 3d.

Consequently, this would change our previous result about the bleaching response. In fact, recruits at 31 °C only exhibited a visible paling because of the faster growth rates and the resultant dilution of pigments, and there was no obvious bleaching either under elevated temperature or temperature fluctuations (Fig. 3d). We will amend this error in the whole manuscript.

References:
1. Babcock R, Mundy C. Coral recruitment: consequences of settlement choice for early growth and survivorship in two scleractinians. J Exp Mar Biol Ecol 206:179–201, 1996.
2. Hughes TP, Jackson JBC. Population dynamics and life histories of foliaceous corals. Ecol Monogr 55:141–66, 1985.
3. Siebeck, U., Marshall, N., Klüter, A., and Hoegh-Guldberg, O.: Monitoring coral bleaching using a colour reference card, Coral Reefs, 25, 453-460, 2006.

[*Comment 5*] *Section 4.4 Please see Buddemeier et al 2008 A modeling tool to evaluate regional coral reef responses to changes in climate and ocean chemistry. Limnology and Oceanography Methods. Particularly their meta-analysis in Figure 2. An alternative explanation may simply be a decreasing slope of the temperature x calcification relationship at higher temperatures as you approach the optimum (Buddemeier Fig. 2), wherein the corals are not calcifying linearly within the temperature fluctuation (i.e. at temperatures above the mean they're not growing much faster and they are growing slower at temperatures below the mean thus resulting in overall decreased calcification in comparison to 31 stable).*

[**Reply**] Thanks for the suggestion on reference and the idea about the non-linear relationship between calcification and temperature. The response of coral skeletal growth to temperature is non-linear and characterized by a parabola whose apogee indicates an optimum and threshold, beyond which the stimulatory impact of temperature will be reversed (Buddemeier et al., 2008; Castillo et al., 2014; Pratchett et al., 2015). Therefore, although the optimal temperature for calcification by *P. damicornis* recruits remains unknown here, it is possible that in the fluctuating 31 °C, recruits may calcify at a slower rate when temperature was above 31 °C during daytime and below 31 °C during night, thus leading to an overall decrease in calcification compared to the constant 31 °C. We will include this alternative explanation in the text.

**Reference:**

1. Buddemeier R W, Jokiel P L, Zimmerman K M, et al. A modeling tool to evaluate regional coral reef responses to changes in climate and ocean chemistry [J]. Limnology & Oceanography Methods, 6(9):395-411, 2008.
2. Pratchett M S, Anderson K D, Hoogenboom M O, et al. Spatial, Temporal and Taxonomic Variation in Coral Growth-Implications for the Structure and Function of Coral Reef Ecosystems [J]. Oceanography & Marine Biology, 53:215-295, 2015.
3. Castillo K D, Ries J B, Bruno J F, et al. The reef-building coral *Siderastrea siderea* exhibits parabolic responses to ocean acidification and warming [J]. Proceedings Biological Sciences, 281(1797), 2014.

**To Referee#2 Dr. E. Rivest**

*General comments: In their manuscript titled "Impact of diurnal temperature fluctuations on larval settlement and growth of the reef coral* Pocillopora damicornis*," the authors present research on an exciting and timely topic – the effect of temperature variability on thermotolerance of two life history stages of a common reef-building coral. The topic is within the scope of the journal and the focus on effects of environmental variability is still novel within the coral field. Unfortunately, I find that this paper is not suitable for publication in its present form. There are several general ways in which this manuscript can be improved.*

**[Reply]** We are deeply grateful for the supreme and considerable efforts of the reviewer to give these valuable and helpful comments. We carefully considered the suggestions and corrections, and made the structure clearer and text more evident to the broad readership of Biogeosciences.

*Reply to specific line-by-line comments:*

**[*Comment 1*]** *The Introduction should include a description of the study species and of their reproduction (brooding) and the fact that the larvae contain symbionts upon release. These are critical pieces of information that the general readership of Biogeosciences will likely not know and are important for properly interpreting the results.*

**[Reply]** Thanks for the suggestion about providing the basic information about the reproductive biology and vertical transmission mode in this coral species. These facts will be added in Introduction. "*P. damicornis* is a widely distributed and major reef-building coral on reef flats in the Indo-Pacific region (Veron 1986). This species planulates almost every month and the release of free-swimming and zooxanthellate planula larvae follows a lunar cycle (Fan et al., 2002)"

References:
1. Veron, J.E.N.. Corals of Australia and the Indo-Pacific. University of Hawaii Press, Honolulu 644 pp, 1986.
2. Fan TY, Li JJ, Ie SX, Fang LS. Lunar periodicity of larval release by pocilloporid corals in southern Taiwan. Zool Stud 41:288–294, 2002

**[*Comment 2*]** *The Methods needs a much better overall description of the experimental design. It is difficult to tell if the spat were from the same or separate trials. Furthermore, the experimental design is flawed because it does not include replication of the treatments and the culturing techniques are not shown to avoid imposing artifacts on the responses of the corals.*

**[Reply]** Sorry for the confusion about the origin of coral spats in Methods. We have revised to make it clear about the two separate experiments. For the settlement assays,

larvae were introduced to the petri-dishes with seawater and a CCA chip to test the effects of temperature treatments on larval settlement. Furthermore, another batch of larvae were transferred to petri-dishes and allowed to settle within 20 hours. Afterwards, these newly settled recruits were randomly assigned to treatment tanks to investigate the temperature effects on the early survival and growth of recruits. These important details will be included in the text.

It was a pity that the experimental design did not include replication and we have explicitly pointed out that limitation and problem in Methods. This problem was addressed by dispensing of larvae/recruits with randomization procedures and controlling other confounding factors such as salinity and light intensity which are of great importance to coral growth (Inoue et al., 2012; Dufault et al., 2013). Secondly, dishes were rotated daily to avoid the potential positional effects within each tank system. All these procedures were performed to ensure similar conditions across treatments except for temperatures during the experiment. Therefore, any differences in the observed responses were due to treatments (Hurlbert 1984; Underwood 1997). Furthermore, this issue was also addressed by carefully examining the significance level of the treatment effects to make sure they were real (All the statistical results will be presented as Tables in Supplement).

References:
1. Dufault A M, Ninokawa A, Bramanti L, et al. The role of light in mediating the effects of ocean acidification on coral calcification [J]. Journal of Experimental Biology, 2013, 216(9):1570-7.
2. Inoue M, Shinmen K, Kawahata H, et al. Estimate of calcification responses to thermal and freshening stresses based on culture experiments with symbiotic and aposymbiotic primary polyps of a coral, *Acropora digitifera* [J]. Global & Planetary Change, 2012, 92-93(s 92–93):1-7.
3. Hurlbert S H. Pseudoreplication and the Design of Ecological Field Experiments [J]. Ecological Monographs, 1984, 54(2):187-211.
4. Underwood AJ. Experiments in ecology: Their logical design and interpretation using analysis of variance. Cambridge University Press, 1997.

[*Comment 3*] *The statistical tests and results need to be fully described. Posthoc analyses are not described. Table(s) with full results of all statistical models should be included, including results of* posthoc *analyses*

[**Reply**] In fact, the statistical results of *post-hoc* analyses have been displayed in the figures and in the text. In Line 235-238, Line 245-246, Line 250-252, Line 259-260 and Line 267-269, results of *post-hoc* analyses of settlement, budding, lateral growth and calcification were described. For instance, when describing the different effects temperature fluctuations on larval settlement and calcification at different mean temperatures, we were just depicting results from the post-hoc analyses. The detailed information of the *post-hoc* analyses will be included as Tables in **Supplement**.

**[*Comment 4*]** *More synthesis and integrative discussion is needed across all the responses measured to inform a broader picture of the implications for the ecology of this coral. The authors need to place their results in the broader context of biogeosciences and coral reef ecology.*

**[Reply]** Thanks for the suggestion about an integrative and broader discussion. To do that, we changed the title of "Conclusions" to "Conclusions and implications", and added a new paragraph after conclusions as follows: "Results of this study suggested that coral larvae subjected to diurnal temperature variations, especially at elevated temperatures, exhibited better settlement competence that those at static thermal treatment. The fluctuating temperatures was favorable for the photo-physiology of endosymbionts and did not put much extra stress on the post-settlement development of coral recruits. Therefore, for corals in highly fluctuating environments, they may have the potential to tolerate and acclimate to the changing seawater temperatures. These findings may also provide clues as to how diverse coral communities can persist and thrive in some thermally variable conditions (Craig et al., 2002; Richards et al., 2015). Furthermore, it is important to note that this study was technically limited to the set of one fluctuating amplitude, and the extent of thermal variance has as much of an impact on fitness as the changes in mean temperature (Vasseur et al. 2014). Given that by now there is still no consensus in the future temperature variability (Burroughs 2007), it will be critical to study the impact of broad ranges of thermal variations which corals may fare in a warming ocean"

References:
1. Craig P, Birkeland C, Belliveau S. High temperatures tolerated by a diverse assemblage of shallow-water corals in American Samoa [J]. Coral Reefs, 20(2):185-189, 2001.
2. Richards Z T, Garcia R A, Wallace C C, et al. A Diverse Assemblage of Reef Corals Thriving in a Dynamic Intertidal Reef Setting (Bonaparte Archipelago, Kimberley, Australia) [J]. PLoS ONE, 10(2):e0117791, 2015.
3. Burroughs WJ (2007) Climate change: a multidisciplinary approach. Cambridge University Press
4. Vasseur DA, DeLong JP, Gilbert B, Greig HS, Harley CDG, McCann KS, Savage V, Tunney TD, O'Connor MI. Increased temperature variation poses a greater risk to species than climate warming. Proceedings of the Royal Society B: Biological Sciences 281, 2014.

**[*Comment 5*]** *L58-59 – "sea surface temperature have increased on average by 0.7 deg C"…since what date? A frame of reference is needed here.*

**[Reply]** Revised as suggested. Now it reads, **"Sea surface temperatures have increased on average by 0.7 °C since preindustrial times (Feely et al., 2009)".**

Reference:

Feely R, Orr J, Fabry V, Kleypas J, Sabine C, Langdon C. Present and future changes in seawater chemistry due to ocean acidification. In: McPherson B, Sundquist E (eds) Carbon sequestration and its role in the global carbon cycle. Geophys Monogr Ser, Vol 183. AGU, p 175−188, 2009.

[*Comment 6*] *L65-70 – it would be good to cite studies that have quantitatively analyzed temperature variability for coral reefs here like Rivest and Gouhier, 2015 and Guadayol et al. 2014*

[**Reply**] Thanks for the suggestion on references. References will be included.

References:
1. Rivest E B, Gouhier T C. Correction: Complex environmental forcing across the biogeographical range of coral populations[J]. PLoS ONE, 10(3): e0121742, 2015.
2. Òscar Guadayol, Silbiger N J, Donahue M J, et al. Patterns in Temporal Variability of Temperature, Oxygen and pH along an Environmental Gradient in a Coral Reef[J]. PLoS ONE, 9(1): e85213, 2014.

[*Comment 7*] *L77-79 – actually, there are a handful of studies (at least 7) that have looked at the effects of temperature variability. I do see that the authors have described the results of a few of these studies in the next paragraph, but they should rephrase this sentence to better define the knowledge gap that their study aims to fill.*

[**Reply**] We have revised this sentence as suggested. Now it reads, "However, only a handful of studies have explored this thermodynamic effect on corals which routinely experience thermal oscillations in nature (e.g., Mayfield et al., 2012; Putnam et al., 2010)"

[*Comment 8*] *L83 – "more suited" is vague and confusing. Please be more specific here.*

[**Reply**] In Longman Dictionary of Contemporary English, "suit" means "be acceptable, suitable or convenient for a particular person or in a particular situation". Therefore, we thought this word choice was proper.

[*Comment 9*] *L84 – "deleterious effects" of what? Diel temperature oscillations?*

[**Reply**] Revised as suggested. Now it reads, "Evidence for the deleterious effects of diel temperature oscillations include the significant reductions in photochemical efficiency, symbiont density and aerobic respiration".

[*Comment 10*] *L86 – "under diel temperature oscillations" compared to what?*

[**Reply**] We added the information as suggested. Now it reads, "under diel temperature

oscillations in comparison to that in constant temperatures".

[***Comment 11***] L90-93 – this statement needs references.

[**Reply**] References will be added as required.

References:
1. Byrne M. Global change ecotoxicology: Identification of early life history bottlenecks in marine invertebrates, variable species responses and variable experimental approaches [J]. Marine Environmental Research, 2012, 76(2):3-15.
2. Keshavmurthy S, Fontana S, Mezaki T, et al. Doors are closing on early development in corals facing climate change [J]. Scientific Reports, 2014, 4:5633.

[***Comment 12***] *L126 – the date of collection of adult corals and the holding conditions of the corals prior to larval release need to be included. The temperature of the water at which the larvae were released should be included.*

[**Reply**] We will include information on date of collection and holding conditions. "Eight *P. damicornis* colonies were collected from 3 m depth on Luhuitou fringing reef on 20 August 2015. Colonies were transported to Tropical Marine Biological Research Station, and placed individually into 20 L flow-through tanks under partially shaded light conditions (noon irradiance, ~300 µmol photons $m^{-2}\,s^{-1}$) and ambient temperature ($28.7 \pm 0.5°C$). The outflow of each tank was passed through a cup fitted with 180 µm mesh on the bottom to trap larvae"

[***Comment 13***] *L129 – "the recruit experiment" – is this the settlement or post-settlement experiment? This should be more clearly defined using a phrase like "to test the effects of xx on yy, larvae were transferred". This is confusing to the reader because the authors have not defined what settlers or recruits are. Remember – the audience is general and interdisciplinary. Or perhaps it would be clearer to describe more generally that the larvae and settlers are being tested in completely separate experiments?*

[**Reply**] Thanks for the suggestion. We revised the title of this Section as "Collection and allocation of coral larvae", and this paragraph as well to clearly define the settlement and recruit experiments as follows "Larvae released from these colonies were collected at 07:00 on 22 August 2015, pooled and haphazardly assigned for the following experiments. For the settlement assays, larvae were transferred to 5.5-cm diameter plastic petri dishes as described below (see Section 2.4). To further test the effects of temperature treatments on the photo-physiology and growth of recruits, another batch of larvae were transferred to 10-cm-diameter petri dishes which were left floating in a flow-through tank. Twenty hours later, 4 dishes with a total of 35–40 newly settled recruits were assigned to each treatment tank. Only recruits that settled individually and at least 1 cm apart from others were selected for the experiment to

avoid possible contact between recruits through growth."

[*Comment 14*] *L130 – were the dishes covered? Did the authors account for/measure effects of evaporation on salinity? Did the authors measure the temperature in the floating dishes during this time? Was there selection that could have influenced the performance of the spat? Again, "spat" is another new synonym used. Please choose one term for the juvenile corals, define it clearly for the reader, and use it consistently throughout the text.*

[**Reply**] All the dishes were covered with close-fitting lids to minimize evaporation while submerged in the tanks. Unfortunately, we did not measure the salinity of seawater within the dish after incubation. Preliminary measurements showed that the temperature difference in seawater temperature between dishes and tanks was less than 0.4 °C and this information will be included in the text. The selection of recruits that were at least 1 cm apart from others was designed to make sure that they will not come into contact through lateral growth. Thanks for the suggestion of the wording for juvenile corals. We will use "recruits" consistently throughout the text.

[*Comment 15*] *L135 – "ambient temperature" where? At the collection site of the adult corals?*
[**Reply**] We revised this sentence to make it clear. Now it reads, **"**The 29 °C treatment, corresponding to the ambient temperature at the collection site of adult *P. damicornis*, was taken to represent the control treatment."

[*Comment 16*] *L153-155 – these are results and should be moved to that section.*

[**Reply**] We moved the information about the temperatures in each treatment to a new section "Treatment conditions" in Results.

[*Comment 17*] *L155 – how was salinity checked?*

[**Reply**] **"**Salinity within each tank was measured using an Orion 013010MD conductivity probe twice a day".

[*Comment 18*] *L159-162 – these are results and should be moved to that section.*

[**Reply**] Information about the light conditions was moved to a new section "Treatment conditions" in Results as suggested.

[*Comment 19*] *L162 – it is a significant limitation that the experiment has no true replication. I understand and empathize with the frustrations of facility and logistical constraints but more justification is needed for the validity of the results. Could the authors repeat the experiment to replicate the results in place of replication during the experiment?*

**[Reply]** We are sorry for this limitation and we have explicitly stated this problem in Methods. To try to eliminate other confounding effects, we randomly allocated coral recruits to each treatment and accurately controlled the salinity and light intensity between treatments. Furthermore, the dishes with recruits were rotated daily within each tank to minimize the potential positional effects. In fact, during the pilot study (as the results presented in Supplement), we failed to manipulate the fluctuating temperature treatments because of a technical problem, and therefore we only reported results of the constant temperature treatments in Supplement. The results of the pilot study were consistent with the later one on the aspect that the growth and development of *P. damicornis* recruits were accelerated at 31 °C, therefore further consolidating our results.

**[*Comment 20*]** *L166 – the title "Settlement assay" makes me think that the authors are going to be testing effects on settlement and is confusing with "preparation of spat" in the title of the last section. Please revise.*

**[Reply]** We changed the structure in Methods and revised the title as suggested to make them clearer and easier to understand. Please see **[Reply]** to **[*Comment 2*] and [*Comment 13*]**.

**[*Comment 21*]** *L168 – is this species of CCA a natural settlement substrate for this species in your location? Please provide additional details here.*

**[Reply]** It has been shown that *P. damicornis* larvae have no specific discrimination for the settlement substrate and it can settle on plastic sheet without the presence of CCA (Hidaka 1985; Lei Jiang personal observation). *Hydrolithon reinboldii* is one of the most abundant CCA species in our study site and juvenile *P. damicornis* in the field are often found adjacent to *H. reinboldii* in our location. Our previous observation found that it was an effective settlement cue for *P. damicornis* larvae. These details will be included in the text.

Reference:
Hidaka M. Tissue compatibility between colonies and between newly settled larvae of *Pocillopora damicornis* [J]. Coral Reefs, 1985, 4(2):111-116.

**[*Comment 22*]** *L170 – did the dishes have lids? Were they sealed in the treatment tank ("submerged")? What was the depth of the water in the dishes? It seems like a very high spat density in a small volume of water. Please provide justification that these are natural and representative settlement conditions for this species.*

**[Reply]** Yes, the petri dishes were covered with lids as they were submerged in the tanks. The depth of the water in each dish was approximately 7 mm. The volume of a single *P. damicornis* larvae ranged from 0.35-0.39 mm$^3$ (Isomura & Nishihira 2001;

Edmunds et al., 2011), and the total volume of 15 larvae was approximately 5.3-5.8 mm$^3$, which only accounted for 0.04% of the total seawater volume (15 ml, 15,000 mm$^3$) in each dish. Furthermore, the larval density in the petri dishes here was 1 larvae per ml, which is representative of that in the literature (e.g., Heyward & Negri 2010; Putnam et al., 2008; Da-Anoy et al., 2017; Harii et al., 2010; Negri et al., 2005).

References:
1. Heyward A J, Negri A P. Plasticity of larval pre-competency in response to temperature: observations on multiple broadcast spawning coral species [J]. Coral Reefs, 2010, 29(3):631-636.
2. Putnam H M, Edmunds P J, Fan T Y. Effect of Temperature on the Settlement Choice and Photophysiology of Larvae From the Reef Coral *Stylophora pistillata* [J]. Biological Bulletin, 2008, 215(2):135-142.
3. Da-Anoy J P, Villanueva R D, Cabaitan P C, et al. Effects of coral extracts on survivorship, swimming behavior, and settlement of *Pocillopora damicornis*, larvae [J]. Journal of Experimental Marine Biology & Ecology, 2017, 486:93-97.
4. Harii S, Yamamoto M, Hoegh-Guldberg O. The relative contribution of dinoflagellate photosynthesis and stored lipids to the survivorship of symbiotic larvae of the reef-building corals [J]. Marine Biology, 2010, 157(6):1215-1224.
5. Isomura N, Nishihira M. Size variation of planulae and its effect on the lifetime of planulae in three Pocilloporid corals [J]. Coral Reefs, 2001, 20(3):309-315.
6. Edmunds P J, Cumbo V, Fan T Y. Effects of temperature on the respiration of brooded larvae from tropical reef corals [J]. Journal of Experimental Biology, 2011, 214(16):2783-90.
7. Negri A, Vollhardt C, Humphrey C, et al. Effects of the herbicide diuron on the early life history stages of coral [J]. Marine Pollution Bulletin, 2005, 51(1):370-383.

[*Comment 23*] *L180 – where did these spat come from? Were they from the "settlement assay" or from "preparation of spat"? Were they kept in the four treatments during this time? I can't interpret the results of these tests without knowing these important details.*

[**Reply**] Sorry again for this structure problem and the confusion it caused. The larvae were randomly used for two separate experiments. Recruits for the post-settlement experiment were all from another batch of larvae which settled on 10-cm-diameter petri-dishes. Also refer to [**Reply**] to [*Comment 2*], [*Comment 13*] **and** [*Comment 20*].

[*Comment 24*] *L194-195 – describe the settings for photography and illumination to allow others to replicate your measurements.*

[**Reply**] ISO setting of the camera was 12800 and the illumination provided while photographing was 35 µmol photons m$^{-2}$ s$^{-1}$. This information will be added in the text.

[*Comment 25*] *L198 – the statistical comparison needs to be described here. What were*

*the controls? Was the bleaching index assessed as relative to corals in the control treatment or was it a comparison of absolute values?*

**[Reply]** Saturation of each coral, a good proxy for chlorophyll/symbiont density (Siebeck et al. 2006), was measured by taking the average value of 30 randomly placed quadrats (100×100 pixels each) on each coral picture using Photoshop's histogram function. The total chlorophyll/symbiont content of each recruit was determined by multiplying the mean saturation by surface area (as measured in Section 2.6 below) to further account for the size difference. Bleaching response was quantified as the reduction in chlorophyll/symbiont content of each recruit relative to the one yielding the maximum value. Therefore, it was just a comparison of the relative values.

Reference:
Siebeck, U., Marshall, N., Klüter, A., and Hoegh-Guldberg, O.: Monitoring coral bleaching using a colour reference card, Coral Reefs, 25, 453-460, 2006.

**[*Comment 26*]** *L201 – which recruits? The ones assessed for bleaching? Different ones?*

**[Reply]** We are sorry for the unclear structure of Methods that made the reviewer feel confused. All the recruits in each treatment were checked daily for their survivorship. At the end of the experiment, recruits were also photographed to assess their surface area and bleaching response. They were the same batch throughout the recruit experiment. To make it clearer, we revised this sentence as "Throughout the recruit experiment, corals from each treatment were checked daily under a dissecting microscope and scored as alive or dead based on the presence of polyp tissue". Also refer to **[Reply]** to **[*Comment 2*], [*Comment 13*], [*Comment 20*]** and **[*Comment 23*]**.

**[*Comment 27*]** *L213 – details of post-hoc analyses need to be included.*

**[Reply]** When main effects were significant ($P < 0.05$), planned multiple comparisons following ANOVAs were conducted using Fisher's LSD test (Day and Quinn, 1989). All the details of *post-hoc* analyses will be included in **Supplement**.

Reference:
Day, R. W., and Quinn, G. P.: Comparisons of treatments after an analysis of variance in ecology, Ecol Monogr, 59, 433-463, 1989.

**[*Comment 28*]** *L229-230 – is this 'normal' settlement behavior for this species? Could it be an artifact of the 'unnatural' settlement conditions?*

**[Reply]** It remains enigmatic whether it was "normal" settlement behavior or it was just an artifact of the "unnatural" settlement conditions. This phenomenon has been confirmed in a wide range of coral species in laboratory (Edmunds et al., 2001; Putnam et al., 2008; Vermeij, 2009; Mizrahi et al., 2014; Richmond, 1985; Denis et al., 2014).

In the discussion part, we presented the possible ecological implications of this kind of larvae according to previous studies (Mizrahi et al., 2014; Richmond, 1985).

References:
1. Edmunds, P., Gates, R., and Gleason, D.: The biology of larvae from the reef coral *Porites astreoides*, and their response to temperature disturbances, Mar. Biol., 139, 981-989, 2001.
2. Putnam H M, Edmunds P J, Fan T Y. Effect of Temperature on the Settlement Choice and Photophysiology of Larvae From the Reef Coral *Stylophora pistillata* [J]. Biological Bulletin, 215(2):135-142, 2008.
3. Vermeij, M. J. A.: Floating corallites: a new ecophenotype in scleractinian corals, Coral Reefs, 28, 987, 2009.
4. Mizrahi, D., Navarrete, S. A., and Flores, A. A. V.: Groups travel further: pelagic metamorphosis and polyp clustering allow higher dispersal potential in sun coral propagules, Coral Reefs, 33, 443-448, 2014.
5. Richmond, R. H.: Reversible metamorphosis in coral planula larvae, Mar. Ecol. Prog. Ser., 22, 181-185, 1985.
6. Denis V, Loubeyres M, Doo S S, et al. Can benthic algae mediate larval behavior and settlement of the coral *Acropora muricata*? [J]. Coral Reefs, 33(2):431-440, 2014.

[**Comment 29**] *L231-235 – since the results were not significant, there are no "distinct" differences. If the interaction is not significant, how can there be significant groupings stated on the figure (2c)?*

[**Reply**] It is certain that "despite a non-significant ANOVA *F*-test there are, in fact, significant differences between at least one set of means among the treatment groups tested which can be ultimately resolved using multiple comparison tests that have more power than the original ANOVA" (Underwood 1997; Dunne 2010; Lesser 2010). Therefore, even if the interaction term was not significant ($P < 0.05$), it did not exclude the possibility of significant groupings. Since the main effect of temperature on settlement was significant, the *post-hoc* analyses did show that the effects of temperature fluctuation were dependent on the mean temperature.

References:
1. Underwood AJ (1997) Experiments in ecology. Cambridge University Press, United Kingdom
2. Dunne R P. Synergy or antagonism—interactions between stressors on coral reefs [J]. Coral Reefs, 2010, 29(1):145-152.
3. Lesser M P. Interactions between stressors on coral reefs: analytical approaches, re-analysis of old data, and different conclusions [J]. Coral Reefs, 2010, 29(3):615-619.

[**Comment 30**] *L237 – "greatly alleviated" is an interpretation and does not belong in*

*the Results section. The phrase "in contrast" is inappropriate here because settlement success was not statistically distinct with that under the fluctuating and constant regimes at 29degC.*

**[Reply]** Revised as suggested in Line 237. About the phrase "in contrast", we thought it was appropriate. It was evident that the temperature fluctuations had different effect on settlement at different mean temperature levels. "Settlement was similar between fluctuating and constant regimes at 29 °C". However, it was not this case at 31 °C. Hence, we used "in contrast" to make a comparison of the effects temperature fluctuations at 29 and 31 °C.

**[*Comment 31*]** *L241 – what were the separate analyses?*

**[Reply]** Separate analyses meant separation of the results by timepoint, i.e., we analyzed the data separately for each timepoint. We revised this sentence as "Separation of the results by time showed that……."

**[*Comment 32*]** *L255 – replace "strongly" with "significantly." Also, the Chi-square test was not listed in the Results section. Please include.*

**[Reply]** The wording is changed as suggested. Moreover, the Chi-square test on the budding state among different treatments was included in Section 2.7 Data analyses in Methods. "Recruits were categorized into 3 states according to the number of polyps: 1-polyp, (2-4)-polyp and (5-6)-polyp. A Chi-square test was used to compare the difference in bud formation among treatments"

**[*Comment 33*]** *L264-267 – again how can the authors claim this if the model was not statistically significant?*

**[Reply]** Again, the reason was that the main effect of temperature was significant, and the temperature fluctuations had different effects on calcification at different mean temperatures as revealed by the *post-hoc* analyses. Please see the explanation in **[Reply]** to **[*Comment 29*]**.

**[*Comment 34*]** *L270 – survival of what?*

**[Reply]** Revised as suggested. Survival of recruits remained >86% in all treatments after 7 days.

**[*Comment 35*]** *L275 – this is the first time Q10 is mentioned. This needs to be included in the methods and defined carefully for the broad readership. Why was Q10 calculated for these results and not the others?*

**[Reply]** Thanks for the suggestion. The definition, calculation formula and the

implications of temperature coefficient Q10 will be added in Methods. Q10 is widely used in temperature experiments to express the sensitivity of metabolism, development and growth to temperature changes (Hochachka & Somero 2002; Rivest & Hofmann 2014; Howe & Marshall 2001). Q10 was calculated using following equation: Q10 = (R2/R1) ^ (10/(T2-T1)), where R is the growth rate at temperature T2 or T1. Q10 values of enzyme-catalyzed reactions often double for the 10 °C increase in temperature. As Q10 is often calculated for respiration, growth and development, here it was calculated for changes in lateral growth, bud development and calcification at two temperatures.

Reference
1. Hochachka, P. W., and Somero, G. N.: Biochemical Adaptation: Mechanism and Process in Physiological Evolution, Oxford University Press, New York, 2002.
2. Rivest E B, Hofmann G E. Responses of the Metabolism of the Larvae of *Pocillopora damicornis* to Ocean Acidification and Warming[J]. PLoS ONE, 2014, 9(4): e96172.
3. Howe, S. A. and A. T. Marshall. Thermal compensation of metabolism in the temperate coral, *Plesiastrea versipora* (Lamarck, 1816). J. Exp. Mar. Biol. Ecol. 259: 231–248, 2001.

[**Comment 36**] *L279 –Based on my interpretation of the data, it was only lower at constant elevated temperatures.*

[**Reply**] Revised as suggested. Now it reads "The pronounced decline in successful settlement at constant 31 °C"

[**Comment 37**] *L282 – "hardly impaired" – too qualitative*

[**Reply**] Revised as suggested. Now it reads, "Interestingly, the transient exposure to 33 °C in variable conditions did not produce the same negative response in larval settlement as the exposure to constant 31 °C; on the contrary, settlement in fluctuating 31 °C was comparable to that in the control treatment".

[**Comment 38**] *L283 – I am having difficulty with the phrase "greatly attenuated the thermal stress on settlement" throughout the manuscript (alleviated, mitigated, tempered….). Because of the lack of replication, it is hard to attribute the responses to thermal stress and constant vs. variable conditions. I think it would be better to say something like "did not produce the same negative response to high temperature as under exposure to constant high temperature." Based on the experimental design, it is impossible to know whether the corals simply experienced less thermal stress overall because they spent some time at temperatures less than 31degC each day or if they responded differently to the high temperature. These mechanistic possibilities should be discussed and phrasing should be more careful.*

[**Reply**] Sorry for this confusion. We have revised the saying in Line 282-283 as

suggested. Please see **[Reply]** to **[*Comment 37*]**. About the replication problem, please see **[Reply]** to **[*Comment 2*]** for explanations and details.

For the word choice of "mitigated, alleviated, tempered.......", we should explain them one by one. For the first one in Line 32, since $Q_m$ was lowered by temperature fluctuations, we revised it as "reduced the maximum excitation pressure". In Line 236-237, we did agree with the reviewer's comment that this was an interpretation and should be stated as facts. Therefore, we revised this sentence as "settlement rates at 31 °C differed significantly among temperature regimes, with settlement being significantly higher at fluctuating conditions and comparable to that in control". For that in Line 282-283, we have revised following the reviewer's suggestion. Please see **[Reply]** to **[*Comment 37*]**. For the one in Line 312, as stated before, there was a significant effect of temperature fluctuation on $Q_m$, and we revised it as "temperature oscillations could relieve the heat stress on corals". For that in Line 423-424, we used the word "tempered" to state that the thermal stress caused by elevated mean temperature (31 °C) on larval settlement was lessened by temperature fluctuations. This was consistent with the meaning of "temper" as "to make something less severe or extreme",

Although, in the fluctuating treatment, corals spent some time at temperatures less than 31 °C each day compared to those in constant 31 °C, that did not mean they experienced less thermal stress overall. Because the experiment was designed to create similar mean temperature values between constant and fluctuating temperature treatments. When determining thermal stress, there must be a reference level. Therefore, relative to control (29 °C in this study), the cumulative thermal stress, as assessed by degree heating days (Maynard et al., 2008), was equivalent for constant and fluctuating 31 °C treatments (corresponding to ~ 2 degree-heating day in the settlement assay). This index is useful in characterizing the experimental heating treatments and facilitating the comparison between temperature treatments (Oliver & Palumbi et al., 2011; Schoepf et al., 2015).

Moreover, we would like to thank the reviewer for the hint that larvae may respond differently to high temperatures. Previous studies have shown that short-term exposure (minutes to hours) of coral larvae to extremely high temperatures (33-37 °C) would enhance the subsequent settlement at lower temperature, suggesting a strong latent effect (Coles 1985; Nozawa & Harrison, 2007). Therefore, another nonexclusive reason for the higher settlement in fluctuating 31 °C may be the 2-h exposure at 33 °C during daytime, thereby exerting a latent effect on settlement at night when the temperature was lowered. This possibility will be added to the Discussion.

References:
1. Maynard, J. A. et al. ReefTemp: An interactive monitoring system for coral bleaching using high-resolution SST and improved stress predictors. Geophys. Res. Lett. 35, L05603, 2008.
2. Schoepf, V., Stat, M., Falter, J. L., and McCulloch, M. T.: Limits to the thermal tolerance of corals adapted to a highly fluctuating, naturally extreme temperature environment. Sci. Rep., 5, 17639, 2015.

3. Oliver, T. A., and Palumbi, S. R.: Do fluctuating temperature environments elevate coral thermal tolerance?. Coral Reefs, 30, 429-440, , 2011

4. Coles SL. The effects of elevated temperature on reef coral planula settlement as related to power station entrainment. In: Proceedings of 5th international coral reef congress. 4:171–176, 1985.

5. Nozawa Y, Harrison P L. Effects of elevated temperature on larval settlement and post-settlement survival in scleractinian corals, *Acropora solitaryensis*, and *Favites chinensis* [J]. Marine Biology, 152(5):1181-1185, 2007.

[**Comment 39**] *L288 – I don't think the authors can say that fluctuating conditions favor settlement because the 29degC constant and fluctuating conditions produced statistically similar settlement rates. Furthermore, when did settlement happen? Did it happen during the daytime when temperatures were higher, or during the nighttime when temperatures were lower? These details could be important for appropriate interpretation of the results.*

[**Reply**] We totally agreed with the reviewer since fluctuating conditions did not impact settlement at 29 °C. However, settlement rates at the mean temperature of 31 °C differed between constant and fluctuating regimes. To address this, we revised this sentence to make it more specific. Now it reads, "whereas settlement may proceed as temperature descends to a more tolerable level at night (30 °C in this study). It is likely that the fluctuating temperature conditions may provide some respite for coral larvae, thereby favoring the settlement at elevated and fluctuating temperatures". For the purpose of not disturbing larvae while handling of petri dishes, we did not monitor settlement at multiple timepoints during incubation. Therefore, we added more discussion here to state this problem and further observations are clearly needed to confirm this hypothesis. The added discussion is as follows, "The use of small petri dishes in the settlement assays restricted the frequent check of larval conditions in order not to disturb larvae. Future studies are needed to regularly observe and establish the dynamics of larval behavior under fluctuating temperatures to confirm this hypothesis".

[**Comment 40**] *L298-301 – what about the desperate larval hypothesis?*

[**Reply**] The desperate larval hypothesis denotes that the non-feeding planktonic larvae become less discriminating in their selection of settlement substrate, i.e., more desperate to settle, as they age and energy reserves run low. The settlement assays only lasted 24 hours, and therefore the desperate larval hypothesis may not fit here.

[**Comment 41**] *L327 – both constant and fluctuating T treatments*

[**Reply**] The +2 °C treatment denoted both the constant and fluctuating 31 °C treatments.

[**Comment 42**] *L340-342 – this sentence needs to be better integrated with the paragraph*

**[Reply]** Thanks for this suggestion and we have revised this paragraph as follows: "Unlike previous work showing the susceptibility of endosymbionts within coral recruits to elevated temperatures (Anlauf et al., 2011; Inoue et al., 2012), there was no significant symbiont loss, i.e., bleaching, for juvenile *P. damicornis* in +2 °C treatments. The resistance of endosymbionts to thermal stress may be linked with the algal type in our study site. *P. damicornis* predominantly harbored *Symbiodinium* clade D in Luhuitou (Zhou 2011), which have been found to be particularly heat resistant (Baker et al., 2004). Further, daytime exposure to high temperatures in fluctuating treatments did not induce the bleaching of juvenile *P. damicornis*. This observation is in stark contrast to that of Putnam and Edmunds (2011) for adult corals. That study found that ephemeral exposure to 30 °C at noon in fluctuating conditions (26–30 °C) elicited a 45% reduction in symbiont density of adult *P. meandrina* compared to those in steady 28 °C, an effect size larger than that arising from continuous exposure to 30 °C (36%). For juvenile corals, the flat structure has been suggested to provide a higher mass transfer capacity to remove reactive oxygen species than the branching and three-dimensional adults (Loya et al., 2001). Hence, the discrepancy between our results and that of Putnam and Edmunds (2011) may, at least partially, be due to the morphology-specific difference in thermal tolerance in corals".

References:
1. Loya, Y., Sakai, K., Yamazato, K., Nakano, Y., Sambali, H., and van Woesik, R.: Coral bleaching: the 538 winners and the losers, Ecol. Lett., 4, 122-131, 2001.
2. Putnam, H. M., and Edmunds, P. J.: The physiological response of reef corals to diel fluctuations in 570 seawater temperature, J. Exp. Mar. Biol. Ecol., 396, 216-223, 2011.
3. Zhou, G.W. Study on diversity of *Symbiodinium* and flexibility in scleractinian coral-algal symbiosis. Ph.D. thesis, Graduate School of Chinese Academy of Sciences, p 127.
4. Baker A C, Starger C J, Mcclanahan T R, et al. Coral reefs: corals' adaptive response to climate change [J]. Nature, 430(7001):741, 2004.

**[*Comment 43*]** *L344 – this section does not mesh well with the rest of the Discussion*

**[Reply]** In fact, section 4.3 was all about the higher growth rate and accelerated development at 31 °C compared to 29 °C, which was an important and independent aspect of this study.

**[*Comment 44*]** *L407-410 – but calcification rates increased under the high temperature treatments…..?*

**[Reply]** Here we are only discussing about the possible explanation for the 20% reduction in calcification at fluctuating 31 °C relative to its constant counterpart. It did

not contradict the fact that recruits calcified faster at higher temperatures.

[*Comment 45*] *L429 – but it was still elevated compared to the 29degC treatments... Figure S1. Panels a and b are not very relevant displays of temperature information for useful interpretation of the experimental design. A plot showing average seasonal daily temperature variability would be more useful. Plot d needs to have an x-axis label.*

[**Reply**] To make it clearer, now it reads "two hours' exposure to 33 °C in fluctuating 31 ° during daytime apparently caused a reduction in calcification compare to that in the constant 31 °C". For Fig S1., S1a did display the seasonal daily average temperatures and daily maximum and minimum values. The bold black line in Fig. s1a shows the daily average temperatures, and the shaded grey area illustrates the daily maximum and minimum temperatures Therefore, it did show the information about the seasonal daily temperature variability. The x-axial label "Date" for S1d was added.

[*Comment 46*] *L116 – Doesn't the dataset go to 2016, not 2015?*

[**Reply**] Change was made as suggested in the text. Now it reads "Seawater temperatures at 3 m depth on Luhuitou fringing reef (18°12′N, 109°28′E) was recorded at 30 min intervals from 2012 to 2016."

[*Comment 47*] *L123 – Should Fig. S1d be cited here instead of S1c?*

[**Reply**] Sorry for this mistake and correction was made accordingly in the text.

[*Comment 48*] *There are consistent errors in grammar and word choice throughout the manuscript. While it does not impede the reader from understanding the scientific content, I advise the authors to carefully copy edit the entire text.*

[**Reply**] We are truly sorry for this problem. With the help of the Editor and all authors, we have tried our best to correct errors in grammar and wording. We will further check the errors and improve the wording. After careful revision, it will be further sent for English editing service.

---

## Author Response (AR1)

**Authors' response to reviewers' comments on the manuscript bg-2017-120 "Impact of diurnal temperature fluctuations on larval settlement and growth of the reef coral *Pocillopora damicornis*" by Lei Jiang et al.**

**To the Editor**

Dear Dr. Christine Klaas,

We would express our sincerest gratitude for your help to correct some errors in the early version of this manuscript, and all the time and efforts it took to develop this manuscript and the review process. We appreciate the constructive comment from the two reviewers. We have carefully considered and incorporated the comments and suggestions from both reviewers and the point-by-point responses are attached as follows. Moreover, we have sent this manuscript for English editing service and amended all the potential grammar errors and wording changes. We are looking forward to receiving your response soon.

Best wishes,

Lei Jiang on behalf of all authors, jianglei12@mails.ucas.ac.cn

**To Referee#1 Dr. D. Barshis**

[*General comments*] *The authors present a comprehensive assessment of the role of diurnally fluctuating temperatures on growth, settlement, and bleaching response of larvae from the coral* Pocillopora damicornis. *The study is quite sound and represents an important contribution to the field. Most coral thermal stress studies use static temperature exposures, hence a movement in the field to more realistic natural thermal profiles is desperately needed. Yet we still lack a fundamental understanding of the different responses of corals to static or variable temperatures in the same study. This research begins to fill in that gap and the manuscript is technically sound and well-presented. There are a few minor comments that should be addressed prior to publication as well as an additional reference that should be integrated into the discussion on growth (see line-by-line comments below). Also, while the writing is generally sound, there are a few instances of misuse of the word "the" and singular/plural errors that may be resolved by additional editing of the language. All in all, I think this is a sound paper that makes an important and needed contribution to the literature.*

[**Reply**] Thanks for the positive comments regarding our manuscript and other insightful and helpful suggestions. We have integrated all the constructive suggestions and further resolved the mistakes about the wording and singular/plural errors through English editing service.

*Reply to specific line-by-line comments:*

[*Comment 1*] *Line 197. Siebeck found brightness and saturation to be indicative of bleaching, why was only saturation used?*

[**Reply**] Work by Siebeck *et al*., 2006 suggested that for pictures of bleached *Pocillopora damicornis*, there were reduced saturation and elevated brightness values. Here, we measured the saturation and brightness values simultaneously and observed the reduction in saturation and increase in brightness (Fig S2). We only presented the saturation value to illustrate the paling of corals at elevated temperatures in the manuscript. We will further include the data on saturation and brightness in **Supplement (Fig. S2)**. Please refer to the Fig. S2 below for further details.

[Figure]

Fig. S2 Photographic metrics for *Pocillopora damicornis* recruits at different temperature treatments.

[**Comment 2**] *Section 2.7 Please specify the software used for statistical tests and*
*copies of code (as supplementary information) if possible.*

[**Reply**] All statistical analyses were performed with STATISTICA version 12.0
(Statsoft). This will be clarified in the text and **Supplement**.

[**Comment 3**] *Line 250-252. Confusing wording. Please clarify that both the elevated*
*31 ℃ stable and 30-33 ℃ fluctuating treatments induced bleaching while the control*
*and 28-31 ℃ fluctuating treatments did not.*

[**Reply**] Revised as suggested. Now it reads "Recruits at 31 ℃ exhibited a paler
appearance than those at 29 ℃, as evidenced by the reduction in saturation and
increase in brightness (Fig. S2). However, bleaching index which accounts for
differences in recruit size, was unaffected by temperature level, regime, or their
interaction (Fig. 3d)". Please see [**Reply**] to [**Comment 4**] below for explanations and
details.

[**Comment 4**] *Lines 327-332. Would add discussion of the increased growth and*
*survival in the higher temps. They may have decreased in color saturation but were*
*not "stressed" according to the other metrics. There could also be a confound wherein*
*a faster growing colony might pale simply because it's growing faster than the*
Symbiodinium *are dividing so it's not losing cells, just diluting pigment. The*
*photographic technique here does not allow for analysis of cell loss and it's unclear*
*over how much area saturation was measured (i.e. how many pixels) and whether it*
*was normalized to surface area or polyp number to account for size differences.*

[**Reply**] The discussion of increased growth and survival at higher temperatures will
be added as follows, "Moreover, recruits with increased growth rates at elevated
temperatures showed higher survivorship, consistent with previous field observations
that survival in early stages of reef corals was strongly dependent on colony size and
growth rates (Babcock and Mundy, 1996; Hughes and Jackson, 1985)".
After carefully examining our data, results totally supported the idea of the
reviewer that coral recruits just became paling because of the faster growth and the
resultant dilution of pigments. We are thankful to the reviewer for pointing out this
puzzle and error. Generally, saturation and brightness of each recruit, were measured
by taking the average value of 30 randomly placed quadrats (100×100 pixels each) on
each coral picture using Photoshop's histogram function (Siebeck et al., 2006). The
quantification of bleaching rates in juvenile corals was quite different from that was
employed for adult branches in Siebeck et al., 2006. For adult branches, the mean
saturation values can be taken as the proxy for symbiont density, however, for the new
recruits here, only the saturation cannot totally reflect the change in symbiont content
in coral holobiont. The bleaching index should consider the change in total content rather than the mean density, because all recruits came from a single coral larva and
recruits had significantly different surface area after exposure to different temperature
conditions. Therefore, to account for the size difference between different treatments,
the total chlorophyll/symbiont content of each recruit was determined by multiplying
the mean saturation by surface area (as measured in Section 2.6). Bleaching response
can be further quantified as the reduction in chlorophyll/symbiont content of each
recruit relative to the one yielding the maximum value. Since we got similar results
from both saturation and brightness measurements, we only presented that results
calculated from saturation in Fig. 3d.
Consequently, this would change our previous result about the bleaching response.
In fact, recruits at 31 °C only exhibited a visible paling because of the faster growth
rates and the resultant dilution of pigments, and there was no obvious bleaching either
under elevated temperature or temperature fluctuations (Fig. 3d). We have amended
this error in the whole manuscript.
References:

1. Babcock R, Mundy C. Coral recruitment: consequences of settlement choice for
early growth and survivorship in two scleractinians. J Exp Mar Biol Ecol
206:179–201, 1996.
2. Hughes TP, Jackson JBC. Population dynamics and life histories of foliaceous
corals. Ecol Monogr 55:141–66, 1985.
3. Siebeck, U., Marshall, N., Klüter, A., and Hoegh-Guldberg, O.: Monitoring coral
bleaching using a colour reference card, Coral Reefs, 25, 453-460, 2006.

**[*Comment 5*]** *Section 4.4 Please see Buddemeier et al 2008 A modeling tool to*
*evaluate regional coral reef responses to changes in climate and ocean chemistry.*
*Limnology and Oceanography Methods. Particularly their meta-analysis in Figure 2.*
*An alternative explanation may simply be a decreasing slope of the temperature x*
*calcification relationship at higher temperatures as you approach the optimum*
*(Buddemeier Fig. 2), wherein the corals are not calcifying linearly within the*
*temperature fluctuation (i.e. at temperatures above the mean they're not growing*
*much faster and they are growing slower at temperatures below the mean thus*
*resulting in overall decreased calcification in comparison to 31 stable).*
**[Reply]** Thanks for the suggestion on reference and the idea about the non-linear
relationship between calcification and temperature. The response of coral skeletal
growth to temperature is non-linear and characterized by a parabola whose apogee
indicates an optimum and threshold, beyond which the stimulatory impact of
temperature will be reversed (Buddemeier et al., 2008; Castillo et al., 2014; Pratchett
et al., 2015). Therefore, although the optimal temperature for calcification by *P.*
*damicornis* recruits remains unknown here, it is possible that in the fluctuating 31 °C,
recruits may calcify at a slower rate when temperature was above 31 °C during
daytime and below 31 °C during night, thus leading to an overall decrease in
calcification compared to the constant 31 °C. We have included this alternative
explanation in the text as follows "The relationship between skeletal growth in corals and temperature is non-linear and characterized by a parabola whose apogee indicated an optimum and threshold, beyond which the stimulatory impact of temperature will be reversed (Buddemeier et al., 2008; Castillo et al., 2014; Inoue et al., 2012; Wŕum et al., 2007). Although the optimal temperature for calcification by P. damicornis recruits remains unknown, it is possible that the recruits exposed to the fluctuating 31 ℃ treatment calcified at a slower rate when the temperature was below 31 ℃ compared to those in the constant 31 ℃. However, given the well-established temperature performance curve for coral calcification (Buddemeier et al., 2008; Wŕum et al., 2007), daytime exposure to temperatures above 32 ℃ would have severely impaired the calcification process, thus leading to an overall decrease in calcification".

**Reference:**
1. Buddemeier R W, Jokiel P L, Zimmerman K M, et al. A modeling tool to evaluate regional coral reef responses to changes in climate and ocean chemistry [J]. Limnology & Oceanography Methods, 6(9):395-411, 2008.
2. Pratchett M S, Anderson K D, Hoogenboom M O, et al. Spatial, Temporal and Taxonomic Variation in Coral Growth-Implications for the Structure and Function of Coral Reef Ecosystems [J]. Oceanography & Marine Biology, 53:215-295, 2015.
3. Castillo K D, Ries J B, Bruno J F, et al. The reef-building coral *Siderastrea siderea* exhibits parabolic responses to ocean acidification and warming [J]. Proceedings Biological Sciences, 281(1797), 2014.

**To Referee#2 Dr. E. Rivest**

*General comments: In their manuscript titled "Impact of diurnal temperature fluctuations on larval settlement and growth of the reef coral* Pocillopora damicornis,*" the authors present research on an exciting and timely topic – the effect of temperature variability on thermotolerance of two life history stages of a common reef-building coral. The topic is within the scope of the journal and the focus on effects of environmental variability is still novel within the coral field. Unfortunately, I find that this paper is not suitable for publication in its present form. There are several general ways in which this manuscript can be improved.*

**[Reply]** We are deeply grateful for the supreme and considerable efforts of the reviewer to give these valuable and helpful comments. We carefully considered the suggestions and corrections, and made the structure clearer and text more evident to the broad readership of Biogeosciences.

*Reply to specific line-by-line comments:*

**[*Comment 1*]** *The Introduction should include a description of the study species and of their reproduction (brooding) and the fact that the larvae contain symbionts upon release. These are critical pieces of information that the general readership of Biogeosciences will likely not know and are important for properly interpreting the results.*

**[Reply]** Thanks for the suggestion about providing the basic information about the reproductive biology and vertical transmission mode in this coral species. These facts will be added in Introduction. "*P. damicornis* is a widely distributed and major reef-building coral on reef flats in the Indo-Pacific region (Veron 1986). This species planulates almost every month and the release of free-swimming and zooxanthellate planula larvae follows a lunar cycle (Fan et al., 2002)"

References:
1. Veron, J.E.N.. Corals of Australia and the Indo-Pacific. University of Hawaii Press, Honolulu 644 pp, 1986.
2. Fan TY, Li JJ, Ie SX, Fang LS. Lunar periodicity of larval release by pocilloporid corals in southern Taiwan. Zool Stud 41:288–294, 2002

**[*Comment 2*]** *The Methods needs a much better overall description of the experimental design. It is difficult to tell if the spat were from the same or separate trials. Furthermore, the experimental design is flawed because it does not include replication of the treatments and the culturing techniques are not shown to avoid imposing artifacts on the responses of the corals.*

**[Reply]** Sorry for the confusion about the origin of coral spats in Methods. We have revised to make it clear about the two separate experiments. For the settlement assays, larvae were introduced to the petri-dishes with seawater and a CCA chip to test the effects of temperature treatments on larval settlement. Furthermore, another batch of larvae were transferred to petri-dishes and allowed to settle within 20 hours. Afterwards, these newly settled recruits were randomly assigned to treatment tanks to investigate the temperature effects on the early survival and growth of recruits. These important details will be included in the text.

It was a pity that the experimental design did not include replication and we have explicitly pointed out that limitation and problem in Methods. This problem was addressed by dispensing of larvae/recruits with randomization procedures and controlling other confounding factors such as salinity and light intensity which are of great importance to coral growth (Inoue et al., 2012; Dufault et al., 2013). Secondly, dishes were rotated daily to avoid the potential positional effects within each tank system. All these procedures were performed to ensure similar conditions across treatments except for temperatures during the experiment, and therefore the observed differences could be attributed to temperature treatments (Hurlbert 1984; Underwood 1997). Furthermore, this issue was also addressed by carefully examining the significance level of the treatment effects to make sure they were real (All the statistical results will be presented as Tables in Supplement).

References:

1. Dufault A M, Ninokawa A, Bramanti L, et al. The role of light in mediating the effects of ocean acidification on coral calcification [J]. Journal of Experimental Biology, 2013, 216(9):1570-7.
2. Inoue M, Shinmen K, Kawahata H, et al. Estimate of calcification responses to thermal and freshening stresses based on culture experiments with symbiotic and aposymbiotic primary polyps of a coral, *Acropora digitifera* [J]. Global & Planetary Change, 2012, 92-93(s 92–93):1-7.
3. Hurlbert S H. Pseudoreplication and the Design of Ecological Field Experiments [J]. Ecological Monographs, 1984, 54(2):187-211.
4. Underwood AJ. Experiments in ecology: Their logical design and interpretation using analysis of variance. Cambridge University Press, 1997.

[**Comment 3**] *The statistical tests and results need to be fully described. Posthoc analyses are not described. Table(s) with full results of all statistical models should be included, including results of* posthoc *analyses*

[**Reply**] In fact, the statistical results of *post-hoc* analyses have been displayed in the figures and in the text. In Line 235-238, Line 245-246, Line 250-252, Line 259-260 and Line 267-269, results of *post-hoc* analyses of settlement, budding, lateral growth and calcification were described. For instance, when describing the different effects temperature fluctuations on larval settlement and calcification at different mean temperatures, we were just depicting results from the post-hoc analyses. The detailed information of the *post-hoc* analyses will be included as Tables in **Supplement**.

[**Comment 4**] *More synthesis and integrative discussion is needed across all the responses measured to inform a broader picture of the implications for the ecology of this coral. The authors need to place their results in the broader context of biogeosciences and coral reef ecology.*

[**Reply**] Thanks for the suggestion about an integrative and broader discussion. However, we feel that results of this study may not be applied broadly, because the temperature variability in marine environment still cannot be accurately predicted by far. To do that, we changed the title of "Conclusions" to "Conclusions and implications", and added a new paragraph after conclusions as follows: "The results of this study suggested that coral larvae subjected to diurnal temperature variations, especially at increased temperature, exhibit better settlement competence than those subjected to static thermal treatment. The fluctuating temperatures were favorable to the photo-physiology of endosymbionts and only had minor effects on post-settlement development of coral recruits. Therefore, for corals in highly fluctuating environments, they may have the potential to tolerate and acclimate to the changing seawater temperatures. These findings may also provide clues as to how diverse coral communities can persist and thrive in some thermally variable conditions (Craig et al., 2001; Richards et al., 2015). It is important to note that this study was technically limited to only one fluctuating amplitude, and the extent of thermal variance has as much of an impact on fitness as the changes in mean temperature (Vasseur et al., 2014). Given that there is currently still no consensus on the future temperature variability (Burroughs, 2007), it will be critical to study the impact of a broad range of thermal variations which corals may fare in a warming ocean."

[**Comment 8**] *L83 – "more suited" is vague and confusing. Please be more specific here.*

[**Reply**] In Longman Dictionary of Contemporary English, "suit" means "be acceptable, suitable or convenient for a particular person or in a particular situation". Therefore, we thought this word choice was proper.

[**Comment 9**] *L84 – "deleterious effects" of what? Diel temperature oscillations?*

[**Reply**] Revised as suggested. Now it reads, "Evidence for the deleterious effects of diel temperature oscillations includes the significant reductions in photochemical efficiency, symbiont density and aerobic respiration".

[**Comment 10**] *L86 – "under diel temperature oscillations" compared to what?*

**[Reply]** We added the information as suggested. Now it reads, "exposed to fluctuating
temperatures compared to those in constant temperatures".
**[*Comment 11*]** L90-93 – this statement needs references.
**[Reply]** References were added as required.

development in corals facing climate change [J]. Scientific Reports, 2014, 4:5633.
**[*Comment 12*]** *L126 – the date of collection of adult corals and the holding*
*conditions of the corals prior to larval release need to be included. The temperature*
*of the water at which the larvae were released should be included.*
**[Reply]** We have included information on date of collection and holding conditions.
"Eight *P. damicornis* colonies were collected from 3 m depth on Luhuitou fringing
reef on 20 August 2015. Colonies were transported to Tropical Marine Biological
Research Station, and placed individually into 20 L flow-through tanks under partially
shaded light conditions (noon irradiance, ~300 µmol photons $m^{-2}$ $s^{-1}$) and ambient
temperature (28.7 ± 0.5 ℃). The outflow of each tank was passed through a cup fitted
with 180 µm mesh on the bottom to trap larvae"
**[*Comment 13*]** *L129 – "the recruit experiment" – is this the settlement or*
*post-settlement experiment? This should be more clearly defined using a phrase like*
*"to test the effects of xx on yy, larvae were transferred". This is confusing to the*
*reader because the authors have not defined what settlers or recruits are. Remember*
*– the audience is general and interdisciplinary. Or perhaps it would be clearer to*
*describe more generally that the larvae and settlers are being tested in completely*
*separate experiments?*
**[Reply]** Thanks for the suggestion. We revised the title of this section as "Collection
and allocation of coral larvae", and this paragraph as well to clearly define the
settlement and recruit experiments as follows "Larvae released from these colonies
were collected at 07:00 on 22 August 2015, pooled and haphazardly assigned for the
following experiments. For the settlement assays, larvae were transferred to 5.5-cm
diameter plastic petri dishes as described below (see Section 2.4). To test the effects
of temperature treatments on the photo-physiology and growth of recruits, another
batch of larvae were transferred to 10-cm-diameter petri dishes which were left
floating in a flow-through tank. Twenty hours later, 4 dishes with a total of 35–40

newly settled recruits were assigned to each treatment tank. Only recruits that settled individually and at least 1 cm apart from others were selected for the experiment to avoid possible contact between recruits through growth."

[*Comment 14*] *L130 – were the dishes covered? Did the authors account for/measure effects of evaporation on salinity? Did the authors measure the temperature in the floating dishes during this time? Was there selection that could have influenced the performance of the spat? Again, "spat" is another new synonym used. Please choose one term for the juvenile corals, define it clearly for the reader, and use it consistently throughout the text.*

[**Reply**] All the dishes were covered with close-fitting lids to minimize evaporation while submerged in the tanks. Unfortunately, we did not measure the salinity of seawater within the dish after incubation. Preliminary measurements showed that the difference in seawater temperature between dishes and tanks was less than 0.4 ℃ and this information will be included in the text. The selection of recruits that were at least 1 cm apart from others was designed to make sure that they will not come into contact through lateral growth. Thanks for the suggestion of the wording for juvenile corals. We will use "recruits" consistently throughout the text.

[*Comment 15*] *L135 – "ambient temperature" where? At the collection site of the adult corals?*

[**Reply**] We revised this sentence to make it clear. Now it reads, **"**The 29 ℃ treatment, corresponding to the ambient temperature at the collection site of adult *P. damicornis*, was taken to represent the control treatment."

[*Comment 16*] *L153-155 – these are results and should be moved to that section.*

[**Reply**] We agreed that these are also results. However, we feel that it is more suitable to present this in Section Materials and methods because they clearly illustrate detailed information on temperature treatments.

[*Comment 17*] *L155 – how was salinity checked?*

[**Reply**] **"**Salinity within each tank was measured using an Orion 013010MD conductivity probe twice a day".

[*Comment 18*] *L159-162 – these are results and should be moved to that section.*

[**Reply**] Although this can also be regarded as results, information about the light conditions was a part of treatment conditions, and therefore we retained this in Materials and methods Section to show that the light conditions were precisely controlled and homogenous across tanks.

[*Comment 19*] *L162 – it is a significant limitation that the experiment has no true*
*replication. I understand and empathize with the frustrations of facility and logistical*
*constraints but more justification is needed for the validity of the results. Could the*
*authors repeat the experiment to replicate the results in place of replication during*
*the experiment?*

[**Reply**] We are sorry for this limitation and we have explicitly stated this problem in
Methods. To try to eliminate other confounding effects, we randomly allocated coral
recruits to each treatment and accurately controlled the salinity and light intensity
between treatments. Furthermore, the dishes with recruits were rotated daily within
each tank to minimize the potential positional effects. In fact, during the pilot study
(as the results presented in Supplement), we failed to manipulate the fluctuating
temperature treatments because of a technical problem, and therefore we only
reported results of the constant temperature treatments in Supplement. The results of
the pilot study were consistent with the later one on the aspect that the growth and
development of *P. damicornis* recruits were accelerated at 31 ℃, therefore further
consolidating our results.

[*Comment 20*] *L166 – the title "Settlement assay" makes me think that the authors*
*are going to be testing effects on settlement and is confusing with "preparation of*
*spat" in the title of the last section. Please revise.*

[**Reply**] We changed the structure in Methods and revised the title as suggested to
make them clearer and easier to understand. Please see [**Reply**] to [*Comment 2*] **and**
[*Comment 13*].

[*Comment 21*] *L168 – is this species of CCA a natural settlement substrate for this*
*species in your location? Please provide additional details here.*

[**Reply**] It has been shown that *P. damicornis* larvae have no specific discrimination
for the settlement substrate and it can settle on plastic sheet without the presence of
CCA (Hidaka 1985; Lei Jiang personal observation). *Hydrolithon reinboldii* is one of
the most abundant CCA species in our study site and juvenile *P. damicornis* in the
field are often found adjacent to *H. reinboldii* in our location. Our previous
observation found that it was an effective settlement cue for *P. damicornis* larvae.
These details are included in the text.

**[*Comment 23*]** *L180 – where did these spat come from? Were they from the "settlement assay" or from "preparation of spat"? Were they kept in the four treatments during this time? I can't interpret the results of these tests without knowing these important details.*

**[Reply]** Sorry again for this structure problem and the confusion it caused. The larvae were mixed and randomly used for two separate experiments. Recruits for the post-settlement experiment were all from another batch of larvae which settled on 10-cm-diameter petri-dishes. Also refer to **[Reply]** to **[*Comment 2*], [*Comment 13*] and [*Comment 20*]**.

**[*Comment 24*]** *L194-195 – describe the settings for photography and illumination to allow others to replicate your measurements.*

**[Reply]** ISO setting of the camera was 12800 and the illumination provided while photographing was 35 μmol photons $m^{-2}$ $s^{-1}$. This information will be added in the text.

**[*Comment 25*]** *L198 – the statistical comparison needs to be described here. What were the controls? Was the bleaching index assessed as relative to corals in the control treatment or was it a comparison of absolute values?*

**[Reply]** Saturation of each coral, a good proxy for chlorophyll/symbiont density (Siebeck et al. 2006), was measured by taking the average value of 30 randomly placed quadrats (100×100 pixels each) on each coral picture using Photoshop's histogram function. The total chlorophyll/symbiont content of each recruit was determined by multiplying the mean saturation by surface area (as measured in Section 2.6 below) to further account for the size difference. Bleaching response was quantified as the reduction in chlorophyll/symbiont content of each recruit relative to the one yielding the maximum value. Therefore, it was just a comparison of the relative values.

[*Comment 29*] *L231-235 – since the results were not significant, there are no*
*"distinct" differences. If the interaction is not significant, how can there be*
*significant groupings stated on the figure (2c)?*
[**Reply**] It is certain that "despite a non-significant ANOVA F-test there are, in fact,
significant differences between at least one set of means among the treatment groups
tested which can be ultimately resolved using multiple comparison tests that have
more power than the original ANOVA" (Underwood 1997; Dunne 2010; Lesser
2010). Therefore, it cannot exclude the possibility of significant groupings though the
interaction term was not significant ($P < 0.05$). Firstly, the main effect of temperature
on settlement was significant, and then the *post-hoc* analyses did show that the effects
of temperature fluctuation were dependent on the mean temperature.

[*Comment 30*] *L237 – "greatly alleviated" is an interpretation and does not belong*
*in the Results section. The phrase "in contrast" is inappropriate here because*
*settlement success was not statistically distinct with that under the fluctuating and*
*constant regimes at 29degC.*
[**Reply**] Revised as suggested. Now it reads "The settlement rate at fluctuating 31 ℃
was comparable to that in the control treatment, and significantly higher than that in
the constant 31 ℃ treatment". About the phrase "in contrast", we thought it was
appropriate. It was evident that the temperature fluctuations had different effect on
settlement at different mean temperature levels. "Settlement was similar between
fluctuating and constant regimes at 29 °C". However, it was not this case at 31 ℃.
Hence, we used "in contrast" to make a comparison of the effects temperature
fluctuations at 29 and 31 ℃.
[*Comment 31*] *L241 – what were the separate analyses?*
[**Reply**] Separate analyses meant separation of the results by timepoint, i.e., we
analyzed the data separately for each timepoint. We revised this sentence as
"Separation of the results by time showed that……."
[*Comment 32*] *L255 – replace "strongly" with "significantly." Also, the Chi-square*
*test was not listed in the Results section. Please include.*
[**Reply**] The wording is changed as suggested. Moreover, the Chi-square test on the
budding state among different treatments was included in Section 2.7 Data analyses in
Methods. "Recruits were divided into 3 categories according to the number of polyps:
1-polyp, (2-4)-polyp and (5-6)-polyp. A Chi-square test was used to compare the
differences in bud formation among treatments."
[*Comment 33*] *L264-267 – again how can the authors claim this if the model was not*
*statistically significant?*
[**Reply**] Again, the reason was that the main effect of temperature was significant, and
the temperature fluctuations had different effects on calcification at different mean temperatures as revealed by the *post-hoc* analyses. Please see the explanation in
**[Reply]** to **[*Comment 29*]**.

**[*Comment 34*]** *L270 – survival of what?*

**[Reply]** Revised as suggested. Survival of recruits remained >86% in all treatments
after 7 days.

**[*Comment 35*]** *L275 – this is the first time Q10 is mentioned. This needs to be*
*included in the methods and defined carefully for the broad readership. Why was Q10*
*calculated for these results and not the others?*

**[Reply]** Thanks for the suggestion. The definition, calculation formula and the
implications of temperature coefficient Q10 will be added in Methods. Q10 is widely
used in temperature experiments to express the sensitivity of metabolism,
development and growth to temperature changes (Hochachka & Somero 2002; Rivest
& Hofmann 2014; Howe & Marshall 2001). Q10 was calculated using following
equation: $Q10 = (R2/R1)$ ^ $(10/(T2-T1))$, where R is the growth rate at temperature T2
or T1. Q10 values of enzyme-catalyzed reactions often double for the 10 ℃ increase
in temperature. As Q10 is often calculated for respiration, growth and development,
here it was calculated for changes in lateral growth, bud development and
calcification at two temperatures.

**[*Comment 36*]** *L279 –Based on my interpretation of the data, it was only lower at*
*constant elevated temperatures.*

**[Reply]** Revised as suggested. Now it reads "The pronounced decline in successful
settlement at constant 31 ℃"

**[*Comment 37*]** *L282 – "hardly impaired" – too qualitative*

**[Reply]** Revised as suggested. Now it reads, "Interestingly, the transient exposure to
33 ℃ in variable conditions did not produce the same negative response on larval
settlement as constant exposure to 31 ℃; on the contrary, coral larvae experiencing diurnal shifts between 30 and 33 ℃ settled at a similar rate to those in the control".

[***Comment 38***] *L283 – I am having difficulty with the phrase "greatly attenuated the*
*thermal stress on settlement" throughout the manuscript (alleviated, mitigated,*
*tempered….). Because of the lack of replication, it is hard to attribute the responses to*
*thermal stress and constant vs. variable conditions. I think it would be better to say*
*something like "did not produce the same negative response to high temperature as*
*under exposure to constant high temperature." Based on the experimental design, it is*
*impossible to know whether the corals simply experienced less thermal stress overall*
*because they spent some time at temperatures less than 31degC each day or if they*
*responded differently to the high temperature. These mechanistic possibilities should*
*be discussed and phrasing should be more careful.*

[**Reply**] Sorry for this confusion. We have revised the saying in Line 282-283 as
suggested. Please see [**Reply**] to [***Comment 37***]. About the replication problem,
please see [**Reply**] to [***Comment 2***] for explanations and details.
For the word choice of "mitigated, alleviated, tempered…….", we should explain
them one by one. For the first one in Line 32, since $Q_m$ was lowered by temperature
fluctuations, we revised it as "reduced the maximum excitation pressure". In Line
236-237, we did agree with the reviewer's comment that this was an interpretation
and should be stated as facts. Therefore, we revised this sentence as "The settlement
rate at fluctuating 31 ℃ was comparable to that in the control treatment and
significantly higher than that in the constant 31 ℃ treatment". For that in Line
282-283, we have revised following the reviewer's suggestion. Please see [**Reply**] to
[***Comment 37***]. For the one in Line 312, as stated before, there was a significant effect
of temperature fluctuation on $Q_m$, and we revised it as "temperature oscillations could
relieve the heat stress on corals". For that in Line 423-424, we used the word
"tempered" to state that the thermal stress caused by elevated mean temperature
(31 ℃) on larval settlement was lessened by temperature fluctuations. This was
consistent with the meaning of "temper" as "to make something less severe or
extreme",
Although, in the fluctuating treatment, corals spent some time at temperatures less
than 31 ℃ each day compared to those in constant 31 ℃, that did not mean they
experienced less thermal stress overall. Because the experiment was designed to
create similar mean temperature values between constant and fluctuating temperature
treatments. When determining thermal stress, there must be a reference level.
Therefore, relative to control (29 ℃ in this study), the cumulative thermal stress, as
assessed by degree heating days (Maynard et al., 2008), was equivalent for constant
and fluctuating 31 ℃ treatments (corresponding to ~ 2 degree-heating day in the
settlement assay). This index is useful in characterizing the experimental heating
treatments and facilitating the comparison between temperature treatments (Oliver &
Palumbi et al., 2011; Schoepf et al., 2015).
Moreover, we would like to thank the reviewer for the hint that larvae may respond
differently to high temperatures. Previous studies have shown that short-term exposure (minutes to hours) of coral larvae to extremely high temperatures (33-37 ℃) would enhance the subsequent settlement at lower temperature, suggesting a strong latent effect (Coles 1985; Nozawa & Harrison, 2007). Therefore, another nonexclusive reason for the higher settlement in fluctuating 31 ℃ may be the 2-h exposure at 33 ℃ during daytime, thereby exerting a latent effect on settlement at night when the temperature was lowered. This possibility is added to the Discussion.

**[Reply]** The desperate larval hypothesis denotes that the non-feeding planktonic larvae become less discriminating in their selection of settlement substrate, i.e., more desperate to settle, as they age and energy reserves run low. The settlement assays only lasted 24 hours, and therefore the desperate larval hypothesis may not fit here.

**[Comment 41]** *L327 – both constant and fluctuating T treatments*

**[Reply]** The +2 ℃ treatment denoted both the constant and fluctuating 31 ℃ treatments.

**[Comment 42]** *L340-342 – this sentence needs to be better integrated with the paragraph*

**[Reply]** Thanks for this suggestion and we have revised this paragraph as follows: "Although juvenile P. damicornis at 31 ℃ exhibited apparent paling appearance compared to those in 29 ℃, loss of symbionts and bleaching were not indicated, as the faster lateral growth at 31 ℃ suggests that the paling is instead the result of pigment dilution due to a larger surface area. This outcome contrasts with previous work showing the sensitivity of endosymbionts within coral recruits to elevated temperatures (Anlauf et al., 2011; Inoue et al., 2012). The lack of bleaching response to elevated temperatures in the current study may be linked to the symbiont type. P. damicornis predominantly harbored Symbiodinium clade D in Luhuitou (Zhou, 2011), which has been found to be particularly thermally tolerant. In addition, the difference in treatment duration could also partially explain these contrasting sensitivities. Albeit ecologically relevant, the exposure duration in this study was much shorter than previous studies (Anlauf et al., 2011; Inoue et al., 2012), therefore resulting in less cumulative stress. It is possible that a loner exposure time may cause similar bleaching responses to those found by other studies.
Further, daytime exposure to high temperatures in fluctuating treatments did not induce significant symbiont loss in juvenile P. damicornis. This observation is in stark contrast to the observations of Putnam and Edmunds (2011) on adult corals. That study found that ephemeral exposure to 30 ℃ at noon in fluctuating conditions (26–30 ℃) elicited a 45% reduction in symbiont density of adult P. meandrina compared to corals at the steady 28 ℃ treatment, a larger effect than that was elicited by continuous exposure to 30 ℃ (36%). The flat structure of juvenile corals has been suggested to provide a higher mass transfer capacity to remove reactive oxygen species than the branching and three-dimensional adults (Loya et al., 2001). Hence, the discrepancy between our results and that of Putnam and Edmunds (2011) may, at least partially, be attributed to the morphology-specific difference in thermal tolerance of juvenile and adult corals".

[**Comment 43**] *L344 – this section does not mesh well with the rest of the Discussion*

[**Reply**] In fact, section 4.3 was all about the higher growth rate and accelerated development at 31 ℃ compared to 29 ℃, which was an important and independent aspect of this study.

[**Comment 44**] *L407-410 – but calcification rates increased under the high temperature treatments…..?*

[**Reply**] Here we are only discussing about the possible explanation for the 20% reduction in calcification at fluctuating 31 ℃ relative to its constant counterpart. It did not contradict the fact that recruits calcified faster at higher temperatures.

[**Comment 45**] *L429 – but it was still elevated compared to the 29degC treatments... Figure S1. Panels a and b are not very relevant displays of temperature information for useful interpretation of the experimental design. A plot showing average seasonal daily temperature variability would be more useful. Plot d needs to have an x-axis label.*

[**Reply**] To make it clearer, now it reads "two hours' exposure to 33 °C during the daytime apparently caused a reduction in calcification compared to constant exposure to 31 ℃". For Fig S1., S1a did display the seasonal daily average temperatures and daily maximum and minimum values. The bold black line in Fig. s1a shows the daily average temperatures, and the shaded grey area illustrates the daily maximum and minimum temperatures Therefore, it did show the information about the seasonal daily temperature variability. The x-axial label "Date" for S1d was added.

[**Comment 46**] *L116 – Doesn't the dataset go to 2016, not 2015?*

[**Reply**] Change was made as suggested in the text. Now it reads "Seawater temperatures at 3 m depth on Luhuitou fringing reef (18°12′N, 109°28′E) was recorded at 30 min intervals from 2012 to 2016."

**[*Comment 47*]** *L123 – Should Fig. S1d be cited here instead of S1c?*

**[Reply]** Sorry for this mistake and correction was made accordingly in the text.

**[*Comment 48*]** *There are consistent errors in grammar and word choice throughout*
*the manuscript. While it does not impede the reader from understanding the scientific*
*content, I advise the authors to carefully copy edit the entire text.*

**[Reply]** We are truly sorry for this problem. With the help of Editor, reviewers and all
authors, we have tried our best to correct errors in grammar and wording. We will
further check the errors and improve the wording. After careful revision, the
manuscript has been sent for English editing and we have carefully copy edited the
entire text.

[revised manuscript text omitted]

---

## Author Response (AR2)

Dear Editor,

We would like to express our sincerest gratitude for the tremendous time and efforts you have devoted to developing this manuscript. We carefully considered the report and made the corrections and changes accordingly. Specifically, about the structure issue in Materials & Methods, we came up with an idea to fix it. We used two section titles for the two separate experiments, i.e., settlement assays and recruit experiment. In turn, we only presented two sections in the Results to make it consistent with the M&M structure. In this way, there will be no misunderstanding of the origin of larvae and measurements. Furthermore, thanks so much for improving the phrasing in this paper. We also added the information about the number of replicate recruits measure for each parameter in the figure legends as required, although some of these details have been mentioned in the main text in M&M. Again thanks for this suggestion which will make the figures more readable and understandable. The point-by-point responses are attached below.

Lei Jiang

On behalf of all authors

[General comments]

Based on the comments of previous reviewers and the answer and the changes to the manuscript I would recommend publication after attending to the following:

- The organization of material and methods section makes it still somewhat difficult to understand which batches where used for which measurement. I would recommend transferring section 2.4 on settlement essays to the end of Materials and Methods (but before section 2.7 on data analyses) as it seems that these essays were carried out separately from the rest of the manipulations. Also, please be more specific on how petri dishes were incubated (as requested by one reviewer) in sections 2.2 and 2.4: were the dishes covered? Were they completely submerged or floating in the incubation tanks?

[Reply] Thanks for the advice about the structuring issue. The transferring of settlement assay part to the end of M&M will disrupt the order in Results as well as that in Discussion. Therefore, we came up with an idea to avoid and fix this problem. We only used two section titles for the two experiments, i.e., larval settlement assay and recruit experiment, and mentioned in the previous section 2.2 that planula larvae collected on 22 August 2015 were pooled and two separate groups of larvae were randomly selected for the following two experiments, respectively. In this way, there will be no difficulty and confusion in understanding of the origin of larvae and recruit in the two distinct experiments.

We added further information about the details of incubation of petri dishes in the two experiments. For the settlement assay, dishes were securely covered with lids to minimize evaporation. Afterwards, they were floated and partially (80%) submerged in the seawater within each tank to ensure temperature control. For the recruit experiment, petri dishes with larvae were left floating on the water surface to allow for larval settlement and recruits preparation. Then, dishes with recruits were assigned to each treatment and placed at the bottom of each tank.

Responses to point-by-point comments:

[Comment 1]- It is my understanding that the manipulations described in sections 2.5 and 2.6 were carried out on the "recruits", hence these sections should be placed after section 2.3. (unless the measurements were carried out on the batches for settlement essays in which case they should be moved after the description of settlement essays).

[Reply] As stated before, we fixed this problem by using only two section titles for the settlement and recruit experiments. Furthermore, we clearly defined the selection of two separate batches of larvae for these two experiments, and moved the description of recruit preparation to the start of section 2.5 recruit experiment. So there will be no misunderstanding of the measurements.

[Comment 2]- Similarly, do the results presented in section 3.1 refer only to the settlement essays? Please specify (maybe adding a sentence at the beginning of the 1st paragraph: "During the settlement essays larval mortality was only observed...").

[Reply] Revised as suggested.

[Comment 3]- Figure 2: How many individuals (n=?) were measured for each treatment? This info could be given in the figure or figure legend.

[Reply] Information was added in the figure legend.

[Comment 4]- Table S3: what do the authors mean by "subject" does it actually correspond to treatment? As in "within treatments" and "between treatments".

[Reply] "Subject" is a term which has been employed in the description and explanation of repeated measures ANOVA. Treatments are between-subject factors and time is treated as the within-subject factor. This expression is widely used in literature.

References:

Ulstrup K E, Berkelmans R, Ralph P J, et al. Variation in bleaching sensitivity of two coral species across a latitudinal gradient on the Great Barrier Reef: the role of zooxanthellae[J]. Marine Ecology Progress, 2006, 314(8):135-148.

Rodrigues L J, Grottoli A G, Lesser M P. Long-term changes in the chlorophyll fluorescence of bleached and recovering corals from Hawaii.[J]. Journal of Experimental Biology, 2008, 211(Pt 15):2502.

Comeau S, Edmunds P J, Spindel N B, et al. Diel pCO2 oscillations modulate the response of the coral Acropora hyacinthus to ocean acidification[J]. Marine Ecology Progress, 2014, 501:99-111.

[Comment 5]- Figure 3: How many individuals (n=?) were measured for each treatment? This info could be given in the figure or in the figure legend.

[Reply] Information was added in the figure legends.

[Comment 6]- Figure 4: How many individuals (n=?) were measured for each treatment? This info could be given in the figure or figure legend.

[Reply] Information was added in the figure legends.

[Comment 7]- Section 3.3 p 14-15 lines 298-305 (2cnd paragraph): the references to the Tables should be corrected (Table S7 and S8 don't exist and Table S6 only contains information on calcification, I imagine the authors refer to Table S5?).

[Reply] Thanks for finding this error and we have checked all the cited supplementary tables in this section.

[Comment 8]- Section 3.3 Line 307: Can the author provide the info on the numbers of recruits? 97% and 86% survivors out of how many?

[Reply] The numbers of recruits at the start of the experiment in each treatment were added in the figure legend.

[Comment 9] p. 3 line 63-64: remove "on coral reef" and striking".

[Reply] Revised as suggested.

[Comment 10] p. 4 line 73: replace "of equivalent..." with "at equivalent..."

[Reply] Revised as suggested.

[Comment 11] p. 4 line 83: replace "is more suite to the fluctuating than to the constant temperatures..." with "is more adapted to fluctuating temperatures..."
[Reply] Revised as suggested.

[Comment 12] p. 4 lines 85-87: replace "Conversely, evidence for the deleterious effects of diel temperature fluctuations includes the significant reductions in photochemical efficiency, symbiont density and aerobic respiration in corals exposed to fluctuating temperatures compared to those in constant temperatures..." with "Conversely, a significant reductions in photochemical efficiency, symbiont density and aerobic respiration were found in corals exposed to fluctuating temperatures..."
[Reply] Thanks for the suggestion and revised as suggested.

[Comment 13] p. 5 line 110: replace "free-swimming and zooxanthellate..." with free-swimming zooxanthellate..."
[Reply] Revised as suggested.

[Comment 14] p. 6 line 126: replace "the diurnal range during summer..." with " The diurnal range in temperature variation during summer..."
[Reply] Revised as suggested.

[Comment 15] p. 6 line 128: replace "temperature began to increase around..." with "temperatures began to rise at around..."
[Reply] Revised as suggested.

[Comment 16] p. 7 section 2.2: reorganize as follows "Eight P. damicornis colonies were collected at a depth of 3 m on 20 August 2015. Colonies were transported to Tropical Marine Biological Research Station, and placed individually into 20 L flow-through tanks at ambient temperature (28.7 _0.5°C) under partially shaded light conditions (noon irradiance, ~300 μmol photons m 2 s 1). The outflow of each tank was passed through a cup fitted with a 180 μm mesh-size net to trap larvae. Larvae were collected at 07:00 on 22 August 2015, pooled and randomly assigned to the following two experiments: 1) To test the effects of temperature treatments on the photo-physiology, bleaching, calcification and growth of recruits, larvae were transferred to 10-cm-diameter petri dishes which were left floating in a flow-through tank. Twenty hours later, 4 dishes with a total of 35–40 newly settled recruits were assigned to each one of the tanks with the four experimental temperature regimes, respectively. Only recruits that settled individually and at least 1 cm apart from others were selected for the experiment in order to avoid possible contact between recruits during growth. Dishes were rotated daily to avoid the potential positional effects within each tank. 2) For the settlement assays, larvae were transferred to 5.5-cm diameter plastic petri dishes as described in Section 2.6."
[Reply] Thanks so much for the restructuring and revised as suggested.

[Comment 17] p. 7-8, section 2.2, 1st paragraph: reorganize as follows:
'The two temperature regimes, constant and fluctuating, were set for the target temperature levels of 29 ℃ and 31 ℃ each. The later temperature value was 2 ℃ above the ambient and 1 ℃ above the bleaching threshold for coral communities on Luhuitou reef (30 ℃, Li et al., 2012), and within the range of projected increases (Bopp et al., 2013).The pattern and range of temperatures in the two fluctuating treatments were based on in situ records obtained during larval release of P. damicornis (Fig. S1d), and the assumption that the predicted 2 ℃ increase in mean temperature would entail a 2 ℃ shift over the whole daily temperature cycle (Burroughs, 2007). The 29 ℃ treatment, corresponding to the ambient temperature at the collection site of adult P. damicornis, was used as the control treatment. "
[Reply] Thanks for the suggestion and revised as suggested.

[Comment 18] p. 8 line 160: replace with "All incubations were carried out in four 40L tanks..."
[Reply] Revised as suggested.

[Comment 19] p. 8, line 162: replace with "Temperature regimes were set using..."
[Reply] Revised as suggested.

[Comment 20] p. 9, line 184-189: replace with "The settlement essays were conducted in 5.5-cm-diameter petri dishes on 22 August 2015 and starting at around 09:00."
[Reply] Revised as suggested.

[Comment 21] p. 12 line 241: replace with "reactions often double with a 10 ℃ rise in temperature."
[Reply] Revised as suggested.

[Comment 22] p. 15 lines 300-305: reorganize as follows:
"The effects of temperature fluctuations on calcification depended on the mean temperature (Fig. 4d), even though the interaction between temperature level and regime was not statistically significant (Table S6.): at 29 ℃, the fluctuating regime had no discernible effect on calcification, while in the fluctuating regime with a mean temperature of 31 ℃ a significant reduction (20%) in calcification was observed when compared to the constant 31 ℃ regime (Table S7.)."
[Reply] Revised as suggested.

[Comment 23] p. 19 line 389: I am not sure what the meaning of "loner" in this sentence is. Do the authors mean "single" or "longer"?
[Reply] Sorry for this mistake and we lost a letter and we actually meant "a longer exposure". We amended this error in the text accordingly.

[Comment 24] p. 20 line 420: replace with "Clearly, thermal tolerance of corals depends on the ambient..."

[Reply] Revised as suggested.

[Comment 25] p. 22 line 458: replace with: "...by a parabola with an optimum..."
[Reply] Revised as suggested.

[Comment 26] p. 22 line 468: replace with: "First, during the warmest part...."
[Reply] Revised as suggested.

[Comment 27] p. 24 lines 507-508: replace with "Therefore, corals in highly fluctuating environments may have the potential..."
[Reply] Revised as suggested.